# Policy Finetuning: Bridging Sample-Efficient Offline and Online Reinforcement Learning

**Tengyang Xie**
UIUC
tx10@illinois.edu

**Nan Jiang**
UIUC
nanjiang@illinois.edu

**Huan Wang**
Salesforce Research
huan.wang@salesforce.com

**Caiming Xiong**
Salesforce Research
cxiong@salesforce.com

**Yu Bai**
Salesforce Research
yu.bai@salesforce.com

## Abstract

Recent theoretical work studies sample-efficient reinforcement learning (RL) extensively in two settings: learning interactively in the environment (online RL), or learning from an offline dataset (offline RL). However, existing algorithms and theories for learning near-optimal policies in these two settings are rather different and disconnected. Towards bridging this gap, this paper initiates the theoretical study of *policy finetuning*, that is, online RL where the learner has additional access to a "reference policy" $\mu$ close to the optimal policy $\pi_\star$ in a certain sense. We consider the policy finetuning problem in episodic Markov Decision Processes (MDPs) with $S$ states, $A$ actions, and horizon length $H$. We first design a sharp *offline reduction* algorithm—which simply executes $\mu$ and runs offline policy optimization on the collected dataset—that finds an $\varepsilon$ near-optimal policy within $\widetilde{O}(H^3 S C^\star / \varepsilon^2)$ episodes, where $C^\star$ is the single-policy concentrability coefficient between $\mu$ and $\pi_\star$. This offline result is the first that matches the sample complexity lower bound in this setting, and resolves a recent open question in offline RL. We then establish an $\Omega(H^3 S \min\{C^\star, A\} / \varepsilon^2)$ sample complexity lower bound for *any* policy finetuning algorithm, including those that can adaptively explore the environment. This implies that—perhaps surprisingly—the optimal policy finetuning algorithm is either offline reduction or a purely online RL algorithm that does not use $\mu$. Finally, we design a new hybrid offline/online algorithm for policy finetuning that achieves better sample complexity than both vanilla offline reduction and purely online RL algorithms, in a relaxed setting where $\mu$ only satisfies concentrability partially up to a certain time step. Overall, our results offer a quantitative understanding on the benefit of a good reference policy, and make a step towards bridging offline and online RL.

## 1 Introduction

Reinforcement learning (RL)—where agents learn to play sequentially in an environment to maximize a cumulative reward function—has achieved great recent success in many artificial intelligence challenges such as video games playing [38, 52], large-scale strategy games (e.g. GO) [44, 45], robotic manipulation [3, 32], behavior learning in social scenarios [8], and more. In many such challenging domains, achieving human-like or superhuman performance requires training the RL agent with millions of samples (steps of acting or game playing) or more. Understanding and improving the sample efficiency of RL algorithms has been a central topic of research.

35th Conference on Neural Information Processing Systems (NeurIPS 2021).

Sample-efficient RL has been studied in a rich body of theoretical work in two main settings: *online RL*, in which the learner has interactive access to the environment and can execute any policy; and *offline RL*, in which the learner only has access to an "offline" dataset collected by executing some (one or many) policies within the environment, and is not allowed to further access the environment. These two settings share some common learning goals such as the sample complexity (number of episodes of playing) for finding the optimal policy. However, existing algorithms and theories in the online and offline setting seem rather different and disconnected—In online RL, state-of-the-art sample-efficient algorithms typically explore the entire environment, e.g. by using optimism to encourage visitation to unseen states and actions [9, 27, 19, 41, 21, 5, 22, 12, 23, 53]. In contrast, offline RL does not allow interactive exploration, and sample-efficient policy optimization algorithms typically focus on optimizing an unbiased (or downward biased) estimator of the value function [39, 48, 4, 40, 10, 56, 35, 58, 25, 42]. It is therefore of interest to ask whether these two types of algorithms and theories can be connected in any way.

Further, on the empirical end, insights and patterns from offline RL often help as well in designing online RL algorithms and improving the sample efficiency in the real world. For example, there are online RL algorithms that alternate between data collection steps using a fixed policy, and policy improvement steps by learning on the collected dataset [20]. The replay buffer in value-based algorithms can also be seen as a local form of offline (off-policy) policy optimization and are often be used in conjunction with optimistic exploration techniques [38, 18, 49]. The prevalence of these algorithms also offers practical motivations for us to look for a more unified understanding of online and offline RL in theory. These reasonings motivate us to ask the following question:

*Can we bridge sample-efficient offline and online RL from a theoretical perspective?*

This paper proposes *policy finetuning*, a new RL setting that investigates the benefit of a good initial policy in reinforcement learning, and encapsulates challenges of both online and offline RL. In the policy finetuning problem, the learner is given interactive access to the environment and asked to learn a near-optimal policy, but in addition has access to a *reference policy* $\mu$ that is good in certain aspects. This setting offers great flexibility for the algorithm design: For example, the algorithm is allowed to either simply collect data from $\mu$ and run any offline policy optimization algorithm on the collected dataset. It is also allowed to play any other policy interactively, including those that adaptively explores the environment. The policy finetuning problem offers a common playground for both offline and online types of algorithms, and has a unified performance metric (sample complexity for finding the near-optimal policy) for comparing their performance.

We study the policy finetuning problem theoretically in finite-horizon Markov Decision Processes (MDPs) with $H$ time steps, $S$ states, and $A$ actions. We summarize our contributions as follows.

- We begin by considering *offline reduction* algorithms which simply collect data using the reference policy $\mu$ and run an offline policy optimization algorithm on the collected dataset. This setting equivalent to offline RL with behavior policy $\mu$, and thus our result translates to a same result for offline RL as well.

  We design an algorithm PEVI-ADV that is able to find an $\varepsilon$-optimal policy (for small $\varepsilon$) within $\widetilde{O}(H^3 S C^\star / \varepsilon^2)$ episodes of play, where $C^\star$ is the *single-policy concentrability* coefficient between $\mu$ and some optimal policy $\pi_\star$ (Section 3). This improves over the best existing offline result by an $H^2$ factor in the same setting and matches the lower bound (up to log factors), thereby resolving the recent open question of [42] on tight offline RL under single-policy concentrability.

- Under the same assumption on $\mu$, we establish an $\Omega(H^3 S \min\{C^\star, A\}/\varepsilon^2)$ sample complexity lower bound for *any* policy finetuning algorithm, including those that adaptively explores the environment (Section 4). This implies that the optimal policy finetuning algorithm is either offline reduction via PEVI-ADV, or a "purely" online RL algorithm from scratch (such as UCBVI), depending on whether $C^\star \leq A$. This comes rather surprising, as it rules out possibilities of combining online exploration and knowledge of $\mu$ to further improve the sample complexity over the aforementioned two baselines.

- Finally, we consider policy finetuning in a more challenging setting where $\mu$ only satisfies concentrability up to a certain time step. We design a "hybrid offline/online" algorithm HOOVI that combines online exploration and offline data collection, and show that it achieves better sample complexity than both vanilla offline reduction and purely online algorithms in certain cases (Section 5). This gives a positive example on when such hybrid algorithm designs are beneficial.

## 1.1 Related work

**Sample-efficient online RL** There is a long line of work on establishing provably sample-efficient online RL algorithms. A major portion of these works is concerned with the tabular setting with finitely many states and actions [9, 27, 19, 5, 11, 2, 22, 63]. For episodic MDPs with inhomogeneous transition functions with $S$ states, and $A$ actions, and horizon length $H$, the optimal sample complexity for finding the $\varepsilon$ near-optimal policy is $\widetilde{O}(H^3 SA/\varepsilon^2)$, achieved by various algorithms such as UCBVI of Azar et al. [5] and UCB-Advantage of Zhang et al. [63]. Our paper adapts the reference-advantage decomposition technique of Zhang et al. [63] to designing sharp offline algorithms. Online RL with with large state/action spaces are also studied by using function approximation in conjunction with structural assumptions on the MDP [23, 61, 62, 1, 41, 21, 47, 53, 57, 14, 24].

**Offline RL** Offline/batch RL studies the case where the agent only has access to an offline dataset obtained by executing a *behavior policy* in the environment. Sample-efficient learning results in offline RL typically work by assuming either sup-concentrability assumptions [39, 48, 4, 40, 15, 51, 10, 56]) or lower bounded exploration constants [58, 59] to ensure the sufficient coverage of offline data over all (relevant) states and actions. However, such strong coverage assumptions can often fail to hold in practice [16]. More recent works address this by using either policy constraint/regularization [16, 35, 29, 55], or the pessimism principle to optimize conservatively on the offline data [30, 60, 28, 25, 59, 42]. The policy-constraint/regularization-based approaches prevent the policy to visit states and actions that has no or low coverage from the offline data. Our proposed offline RL algorithm PEVI-ADV (Algorithm 1) is inspired by the pessimistic value iteration algorithms of [25, 42] and achieves an improved sample complexity over these work under the same single-policy concentrability assumption on the behavior policy.

**Bridging online and offline RL** Kalashnikov et al. [26] observed empirically that the performance of policies trained purely from offline data can be improved considerably by a small amount of additional online fine-tuning. A recent line of work studied low switching cost RL [6, 63, 17, 54]—which forbits online RL algorithms from switching its policy too often—as an interpolation between the online and offline settings. The same problem is also studied empirically as deployment-efficient RL [36, 46]. While we also attempt to bridge online and offline RL, our work differs from this line in that our policy finetuning setting allows a direct comparison between "fully offline" and "fully online" algorithms, whereas the low switching cost setting prohibits fully online algorithms.

## 2 Preliminaries

**Markov Decision Processes** In this paper, we consider episodic Markov decision processes (MDPs) with time-inhomogeneous transitions, specified by $M = (\mathcal{S}, \mathcal{A}, H, \mathbb{P}, r)$, where $\mathcal{S}$ is the state space, $\mathcal{A}$ is the action space, $H$ is the horizon length, $\mathbb{P} = \{\mathbb{P}_h\}_{h=1}^H$ where $\mathbb{P}_h(\cdot|s,a) \in \Delta_{\mathcal{S}}$ is the transition probabilities at step $h$, and $r = \{r_h : \mathcal{S} \times \mathcal{A} \to [0,1]\}_{h=1}^H$ are the deterministic[1] reward functions at time step $h \in [H]$. Without loss of generality, we assume that the initial state $s_1$ is deterministic[2].

**Policies, value functions, visitation distributions** A policy $\pi = \{\pi_h(\cdot|s)\}_{h \in [H], s \in \mathcal{S}}$ consists of distributions $\pi_h(\cdot|s) \in \Delta_{\mathcal{A}}$. We use $\mathbb{E}_\pi[\cdot]$ to denote the expectation with respect to the random trajectory induced by $\pi$ in the MDP $M$, that is, $(s_1, a_1, r_1, s_2, a_2, r_2, \ldots, s_H, a_H, r_H)$, where $a_h = \pi_h(s_h)$, $r_h = r_h(s_h, a_h)$, $s_{h+1} \sim \mathbb{P}_h(\cdot|s_h, a_h)$. For each policy $\pi$, let $V_h^\pi : \mathcal{S} \to \mathbb{R}$ and $Q_h^\pi : \mathcal{S} \times \mathcal{A} \to \mathbb{R}$ denote its value functions and Q functions at each time step $h \in [H]$, that is,

$$V_h^\pi(s) := \mathbb{E}_\pi\left[\sum_{h'=h}^H r_{h'}(s_{h'}, a_{h'}) \bigg| s_h = s\right], \ Q_h^\pi(s,a) := \mathbb{E}_\pi\left[\sum_{h'=h}^H r_{h'}(s_{h'}, a_{h'}) \bigg| s_h = s, a_h = a\right].$$

The operators $\mathbb{P}_h$ and $\mathbb{V}_h$ are defined as $[\mathbb{P}_h V_{h+1}](s,a) := \mathbb{E}[V_{h+1}(s')|s_h = s, a_h = a]$ and $[\mathbb{V}_h V_{h+1}](s,a) := \mathrm{Var}[V_{h+1}(s')|s_h = s, a_h = a]$ for any value function $V_{h+1}$ at time step $h+1$.

---

[1] While we assume deterministic rewards for simplicity, our results can be straightforwardly generalized to stochastic rewards, as the major difficulty is in learning the transitions rather than learning the rewards.

[2] Any MDP with stochastic $s_1$ is equivalent to an MDP with deterministic by creating a dummy initial state $s_0$ and increasing the horizon by 1.

We also use $\widehat{\mathbb{P}}_h$ and $\widehat{V}_h$ to denote empirical versions of these operators building on estimated models (which will be clear in the context).

We use $\pi_\star := \arg\max_\pi V_1^\pi(s_1)$ to denote any optimal policy, and $V_h^\star := V_h^{\pi_\star}$ and $Q_h^\star := Q_h^{\pi_\star}$ to denote the value function and Q function of $\pi^\star$ at all $h \in [H]$. Throughout this paper, our learning goal is to find an near-optimal policy $\widehat{\pi}$ such that $V_1^\star(s_1) - V_1^{\widehat{\pi}}(s_1) \le \varepsilon$.

Finally, we let $d_h^\pi$ denote the state(-action) visitation distributions of $\pi$ at time step $h \in [H]$:

$$d_h^\pi(s) := \mathbb{P}(s_h = s|\pi), \quad \text{and} \quad d_h^\pi(s,a) := \mathbb{P}(s_h = s, a_h = a|\pi).$$

**Miscellaneous**   We use standard $O(\cdot)$ and $\Omega(\cdot)$ notation: $A = O(B)$ is defined as $A \le CB$ for some absolute constant $C > 0$ (and similarly for $\Omega$). The tilded notation $A = \widetilde{O}(B)$ denotes $A \le CL \cdot B$ where $L$ is a poly-logarithmic factor of problem parameters.

### 2.1   Policy Finetuning

We now introduce the setting of *policy finetuning*. A policy finetuning problem consists of an MDP $M$ and a *reference policy* $\mu$. During the learning stage, the learner can perform the following two types of moves:

(a) Play an episode in the MDP $M$ using any policy (i.e. learner has online interactive access to $M$).

(b) Access the values of the reference policy $\mu_h(a|s)$ for all $(h, s, a)$. For example, the learner can use it to sample actions $a \sim \mu_h(\cdot|s)$ for any $h, s$ for arbitrarily many times during learning.

The goal of the learner is to output $\varepsilon$ near-optimal policy $\widehat{\pi}$ within as few episodes of play (within the MDP) as possible.

A unique feature about the policy finetuning setting is that it allows both *online interactive plays* via any online RL algorithm (not necessarily using $\mu$), as well as *offline reduction* which simply collects data by executing the reference policy $\mu$ and do anything with the collected dataset. In particular, this means that any algorithm for offline policy optimization (based on offline datasets) also gives an algorithm for policy finetuning via this offline reduction. Therefore, policy finetuning offers a common playground for both online and offline type algorithms with a unified learning goal.

**Assumption on reference policy**   Throughout most of this paper (except for Section 5), we consider the following assumption on the reference policy $\mu$.

**Assumption A** (Single-policy concentrability)**.** *The reference policy $\mu$ satisfies that*

$$\max_{h\in[H],(s,a)\in\mathcal{S}\times\mathcal{A}} \frac{d_h^{\pi_\star}(s,a)}{d_h^\mu(s,a)} \le C^\star$$

*(with the convention $0/0 = 0$) for some* deterministic *optimal policy $\pi_\star$ and constant $C^\star \ge 1$.*

The single-policy concentrability characterizes the distance between the visitation distributions of the reference policy $\mu$ and some optimal policy $\pi^\star$. This assumption is considered in the recent work of Rashidinejad et al. [42] on offline RL and is more relaxed than previously assumed concentrability assumptions which typically requires the supremum concentrability against all possible $\pi$'s to be bounded [10]. We consider this assumption as it both allows efficient offline RL algorithms [42], and is perhaps also a sensible measure of quality for the reference policy in policy finetuning.

## 3   Sharp offline learning via reference-advantage decomposition

We begin by investigating the sharpest sample complexity for policy finetuning via the offline reduction approach. This requires us to design sharp offline RL algorithms that run on the dataset $\mathcal{D}$ collected by executing $\mu$. We emphasize that this is both an interesting offline RL question on its own right, and also important for our later discussions on lower bounds and other algorithms for policy finetuning, as the sharpest sample complexity via offline reduction provides a solid baseline.

**Warm-up: VI-LCB**  As a warm-up, we first show that a finite-horizon variant of the VI-LCB (Value Iteration with Lower Confidence Bounds) algorithm of Rashidinejad et al. [42] achieves sample complexity $\widetilde{O}(H^5 S C^\star / \varepsilon^2)$ for finding an $\varepsilon$ near-optimal policy. This result is similar to the $\widetilde{O}(S C^\star / (1-\gamma)^5 \varepsilon^2)$ guarantee[3] for the original VI-LCB in infinite-horizon discounted MDPs [42, Theorem 6]. The main ingredients of our VI-LCB algorithm is a pessimistic value iteration procedure in which we perform value iteration on the empirical model estimated from the dataset $\mathcal{D}$, along with a negative Hoeffding bonus term to impose pessimism. Due to space constraints, the algorithm description (Algorithm 3) and the proof of Theorem 1 are deferred to Appendix B.

**Theorem 1** (VI-LCB for finite-horizon MDPs). *Suppose the reference policy $\mu$ satisfies the single-policy concentrability (Assumption A). Then with probability at least $1 - \delta$,* VI-LCB *(Algorithm 3) outputs a policy $\widehat{\pi}$ and value estimate $\widehat{V}$ such that*

(a) $\max_{h \in [H]} \sum_{s \in \mathcal{S}} d_h^{\pi^\star}(s)(V_h^\star(s) - \widehat{V}_h(s)) \leq \varepsilon$,

(b) $V_1^\star(s_1) - V_1^{\widehat{\pi}}(s_1) \leq \varepsilon$,

*within $n = \widetilde{O}\big(H^5 S C^\star / \varepsilon^2\big)$ episodes.*

Theorem 1 serves two main purposes. First, the $\widetilde{O}(H^5 S C^\star / \varepsilon^2)$ sample complexity asserted in Theorem 1(b) provides a first result for offline RL (and offline reduction for policy finetuning) under single-policy concentrability in finite-horizon MDPs. Second, the value estimation bound in Theorem 1(a) shows that the estimated value function $\widehat{V}_h(s)$ provided by VI-LCB is close to the optimal value $V_h^\star(s)$ *at every step $h \in [H]$*, in terms of the weighted average with $d_h^{\pi^\star}(s)$. Our next algorithm PEVI-ADV builds on this property so that VI-LCB can be used as a "warm-up" learning procedure that provides a high-quality value estimate.

**Sharp offline learning via reference-advantage decomposition**  We now design a new sharp algorithm PEVI-ADV which achieves an improved $\widetilde{O}(H^3 S C^\star / \varepsilon^2)$ sample complexity (for small enough $\varepsilon$). This improves over VI-LCB by $\widetilde{O}(H^2)$ and is the first algorithm that matches the sample complexity lower bound. PEVI-ADV adds two new ingredients over VI-LCB in order to achieve the $\widetilde{O}(H^2)$ improvement:

1. We replace the Hoeffding-style bonus in VI-LCB with a Bernstein-style bonus. This shaves off one $H$ factor in the sample complexity via the total variance property (Lemma C.4).

2. Both VI-LCB and our PEVI-ADV use data splitting to make sure that the estimated value $\widehat{V}_{h+1}$ and empirical transitions $\widehat{\mathbb{P}}_h$ are estimated using different subsets of $\mathcal{D}$, this yields conditional independence that is required in bounding concentration terms of the form $(\widehat{\mathbb{P}}_h - \mathbb{P}_h)\widehat{V}_{h+1}$. However, applied naively, this data splitting induces one undesired $H$ factor in the sample complexity as we need to split $\mathcal{D}$ into $H$ folds and thus each $\mathbb{P}_h$ is estimated using only $n/H$ episodes of data.

   As a technical crux of this algorithm, we overcome this issue by adapting the *reference-advantage decomposition* technique of Zhang et al. [63]. This technique proposes to learn an initial reference value function $\widehat{V}^{\mathrm{ref}}$ of good quality in a certain sense, and then performing the following type of approximate value iteration (using the right-hand side as the algorithm update):

$$\mathbb{P}_h \widehat{V}_{h+1} \approx \widehat{\mathbb{P}}_{h,0} \widehat{V}_{h+1}^{\mathrm{ref}} + \widehat{\mathbb{P}}_{h,1}\left(\widehat{V}_{h+1} - \widehat{V}_{h+1}^{\mathrm{ref}}\right).$$

   Above, $\widehat{V}_{h+1}$, $\widehat{\mathbb{P}}_{h,0}$, and $\widehat{\mathbb{P}}_{h,1}$ are estimated on three disjoint subsets of the data. The advantage of this approach is that, due to this new independence structure, $\widehat{\mathbb{P}}_{h,0}$ for different $h \in [H]$ can be estimated on the same set of trajectories without $H$-fold splitting, which shaves off the $H$ factor within this part. On the other hand, estimating $\widehat{\mathbb{P}}_{h,1}$ still requires $H$-fold splitting, yet this would not hurt the sample complexity if the magnitude of $(\widehat{V}_{h+1} - \widehat{V}_{h+1}^{\mathrm{ref}})$ is much smaller than its naive upper bound $O(H)$—we show this can be achieved by using VI-LCB to learn $\widehat{V}^{\mathrm{ref}}$.

---

[3][42] can achieve a faster rate in case $C^\star \leq 1 + \widetilde{O}(1/N)$. However, we focus on the case $C^\star = 1 + \Theta(1)$ where the guarantee of Rashidinejad et al. [42] is $\widetilde{O}(S C^\star / (1-\gamma)^5 \varepsilon^2)$.

---

**Algorithm 1** Pessimistic Value Iteration with Reference-Advantage Decomposition (PEVI-ADV)

---

**Require:** Dataset $\mathcal{D} = \left\{ (s_1^{(i)}, a_1^{(i)}, r_1^{(i)}, \ldots, s_H^{(i)}, a_H^{(i)}, r_H^{(i)}) \right\}_{i=1}^{n}$ collected by executing $\mu$ in $M$.

1: Split the dataset $\mathcal{D}$ into $\mathcal{D}_{\text{ref}}, \mathcal{D}_0$ and $\{\mathcal{D}_{h,1}\}_{h=1}^{H}$ uniformly at random:

$$n_{\text{ref}} := |\mathcal{D}_{\text{ref}}| = n/3, \ \ n_0 := |\mathcal{D}_0| = n/3, \ \ n_{1,h} := |\mathcal{D}_{h,1}| := n/(3H) \ \ (n_1 := n/3).$$

2: Learn a reference value function $\widehat{V}^{\text{ref}} \leftarrow$ VI-LCB($\mathcal{D}_{\text{ref}}$) via VI-LCB (Algorithm 3).
3: Let $N_{h,0}(s,a)$ and $N_{h,0}(s,a,s')$ denote the visitation count of $(s,a)$ and $(s,a,s')$ at step $h$ within dataset $\mathcal{D}_0$. Construct empirical model estimates:

$$\widehat{\mathbb{P}}_{h,0}(s'|s,a) \leftarrow \frac{N_{h,0}(s,a,s')}{N_{h,0}(s,a) \vee 1}, \ \ \text{and} \ \ \widehat{r}_{h,0}(s,a) \leftarrow r_h(s,a)\mathbb{1}\left\{N_{h,0}(s,a) \geq 1\right\}.$$

Similarly define $N_{h,1}(s,a)$, $N_{h,1}(s,a,s')$, $(\widehat{r}_{h,1}, \widehat{\mathbb{P}}_{h,1})$ for all $h \in [H]$ based on dataset $\mathcal{D}_{h,1}$.

4: Set $b_{h,0}(s,a) \leftarrow c \cdot \left( \sqrt{\frac{[\widehat{\mathbb{V}}_{h,0}\widehat{V}_{h+1}^{\text{ref}}](s,a)\iota}{N_{h,0}(s,a) \vee 1}} + \frac{H\iota}{N_{h,0}(s,a) \vee 1} \right)$ for all $(h,s,a)$, where $\iota := \log(HSA/\delta)$.

5: Set $\widehat{V}_{H+1}(s) \leftarrow 0$ for all $s \in \mathcal{S}$.
6: **for** $h = H, \ldots, 1$ **do**

7:   Set $b_{h,1}(s,a) \leftarrow c \cdot \left( \sqrt{\frac{[\widehat{\mathbb{V}}_{h,1}(\widehat{V}_{h+1} - \widehat{V}_{h+1}^{\text{ref}})](s,a)\iota}{N_{h,1}(s,a) \vee 1}} + \frac{H\iota}{N_{h,1}(s,a) \vee 1} \right)$.

8:   Perform pessimistic value update for all $(s,a)$:

$$\widehat{Q}_h(s,a) \leftarrow \widehat{r}_{h,0}(s,a) + \left[\widehat{\mathbb{P}}_{h,0}\widehat{V}_{h+1}^{\text{ref}}\right](s,a) - b_{h,0}(s,a) + \left[\widehat{\mathbb{P}}_{h,1}(\widehat{V}_{h+1} - \widehat{V}_{h+1}^{\text{ref}})\right](s,a) - b_{h,1}(s,a);$$

$$\widehat{V}_h(s) \leftarrow \left[\max_a \widehat{Q}_h(s,a)\right] \vee 0.$$

9:   Set $\widehat{\pi}_h(s) \leftarrow \arg\max_a \widehat{Q}_h(s,a)$ for all $s \in \mathcal{S}$.
10: **end for**
11: **return** Policy $\widehat{\pi} = \{\widehat{\pi}_h\}_{h \in [H]}$.

---

We instantiate this plan by carefully using VI-LCB to learn the reference value function $\widehat{V}^{\text{ref}}$, combined with tight Bernstein bonuses, to shave off another $H$ factor in the sample complexity. The full PEVI-ADV algorithm is provided in Algorithm 1. We now present its guarantee in the following theorem. The proof can be found in Appendix C.

**Theorem 2** (Sharp offline learning via PEVI-ADV). *Suppose the reference policy $\mu$ satisfies the single-policy concentrability (Assumption A). Then with probability at least $1 - \delta$, PEVI-ADV (Algorithm 1) outputs a policy $\widehat{\pi}$ and value estimate $\widehat{V}$ such that*

*(a)* $\max_{h \in [H]} \sum_{s \in \mathcal{S}} d_h^{\pi^\star}(s)(V_h^\star(s) - \widehat{V}_h(s)) \leq \varepsilon,$

*(b)* $V_1^\star(s_1) - V_1^{\widehat{\pi}}(s_1) \leq \varepsilon,$

*within* $n = \widetilde{O}\big(H^3 SC^\star/\varepsilon^2 + H^{5.5}SC^\star/\varepsilon\big)$ *episodes.*

**Near-optimal offline RL under single-policy concentrability**  For small enough $\varepsilon \leq H^{-2.5}$, Theorem 2 achieves $\widetilde{O}(H^3 SC^\star/\varepsilon^2)$ sample complexity for finding the $\varepsilon$ near-optimal policy from the offlien dataset $\mathcal{D}$. This is the first cubic horizon dependence for offline RL under single-policy concentrability, which improves over recent works [25, 42] in this setting and resolves the open question of [42]. For $C^\star \geq 2$, our sample complexity further matches the information-theoretical lower bound $\Omega(H^3 SC^\star/\varepsilon^2)$ up to log factors[4]. We remark that tight hoizron dependence has also been achieved in several recent works offline RL [58, 59, 43] which are however quite different from (and do not imply) ours in both the assumptions (on the behavior policy) and the analyses.

---

[4]This lower bound can be adapted directly from a $\Omega(SC^\star/(1-\gamma)^3\varepsilon^2)$ lower bound of [42, Theorem 7].

# 4 Lower bound for policy finetuning

We now switch gears to considering the policy finetuning problem with any algorithm, not necessarily restricted to the offline reduction approach.

**Two baselines: offline reduction & purely online RL** A first observation is that naive offline reduction is already a strong baseline for policy finetuning, by our Theorem 2: Our PEVI-ADV algorithm only collects data with $\mu$ and does not do any online exploration, yet achieves a sharp $\widetilde{O}(H^3 SC^\star/\varepsilon^2)$ sample complexity for finding a near-optimal policy.

On the other hand, as the policy finetuning setting allows online interaction, *purely online RL* is another baseline algorithm: Simply run any sample-efficient online RL algorithm (which typically uses optimism to encourage exploration) from scratch, and disregard the reference policy $\mu$. Using any sharp online RL algorithm such as UCBVI [5], this approach can find an $\varepsilon$ near-optimal policy within $\widetilde{O}(H^3 SA/\varepsilon^2)$ episodes of play. Note that whether this is advantageous over the offline reduction boils down to the comparison between $C^\star$ and $A$, which makes sense intuitively. For example, $C^\star \leq o(A)$ means that $\mu$ is perhaps close enough to $\pi_\star$ so that collecting data from $\mu$ and run offline policy optimization is a stronger algorithm than exploring from scratch.

Given these two baselines, it is natural to ask whether there exists an algorithm that improves over both — Can we design an algorithm that performs some amount of optimistic exploration, yet also utilizes the knowledge of $\mu$, so as to achieve a better rate than both offline reduction and purely online RL? In this section, we provide an information-theoretic lower bound showing that, perhaps surprisingly, the answer is negative: there is an $\Omega(H^3 S \min\{C^\star, A\}/\varepsilon^2)$ sample complexity lower bound for any policy finetuning algorithm, if we still assume that $\mu$ satisfies $C^\star$ single-policy concentrability.

**Lower bound** To formally state our lower bound, we define the class of problems

$$\mathcal{M}_{C^\star} := \left\{ (M, \mu) : \text{ Exists deterministic } \pi_\star \text{ of } M \text{ such that } \sup_{h,s,a} \frac{d_h^{\pi_\star}(s,a)}{d_h^\mu(s,a)} \leq C^\star \right\}. \quad (1)$$

We recall that a policy finetuning algorithm for problem $(M, \mu)$ is defined as any algorithm that can play in the MDP $M$ for $n$ episodes, has full knowledge of the reference policy $\mu$, and outputs a policy $\widehat{\pi}$ after playing in the MDP.

With these definitions ready, we now state our lower bound for policy finetuning. The proof of Theorem 3 can be found in Appendix D.

**Theorem 3** (Lower bound for policy finetuning). *Suppose $S, H \geq 3$, $A \geq 2$, $C^\star \geq 2$. Then, there exists an absolute constant $c_0 > 0$ such that for any $\varepsilon \leq 1/12$ and any online finetuning algorithm that outputs a policy $\widehat{\pi}$, if the number of episodes*

$$n \leq c_0 \cdot H^3 S \min\{C^\star, A\}/\varepsilon^2,$$

*then there exists a problem instance $(M, \mu) \in \mathcal{M}_{C^\star}$ on which the algorithm suffers from $\varepsilon$-suboptimality:*

$$\mathbb{E}_M\left[ V_{1,M}^\star - V_{1,M}^{\widehat{\pi}} \right] \geq \varepsilon,$$

*where the expectation $\mathbb{E}_M$ is w.r.t. the randomness during the algorithm execution within MDP $M$.*

**Either offline reduction or purely online is optimal** Theroem 3 shows that any policy finetuning algorithm needs to play at least $\Omega(H^3 S \min\{C^\star, A\}/\varepsilon^2)$ episodes in order to find an $\varepsilon$ near-optimal policy. Crucially, this implies that either a sharp offline reduction (e.g. our PEVI-ADV algorithm) or purely online RL matches the lower bound (up to log), depending on whether $C^\star \lesssim A$. In other words, if we have the knowledge of whether $C^\star \leq A$, choosing the right one of these two baseline algorithms will yield the optimal sample complexity. Perhaps surprisingly, this rules out the possibility of designing any algorithm "in between" that combines online exploration and knowledge of $\mu$ to improve the sample complexity, at least in the worst-case over all problems in $\mathcal{M}_{C^\star}$. We argue that this "no algorithm in between" phenomenon may be due to the single-policy concentrability assumption being too strong such that offline reduction already achieves a rather competitive sample

---

**Algorithm 2** Hybrid Offline/Online Value Iteration (HOOVI)

---

**Require:** MDP $M$, reference policy $\mu$.

1: # Stage 1: Learn step $h_\star + 1 : H$ via optimistic online exploration
2: **for** Episode $k = 1, \ldots, n_{\mathrm{UCB}} = n/2$ **do**
3:    Receive initial state $s_1$ and play with policy $\mu$ up to step $h_\star$. Arrive at state $s_{h_\star+1}$.
4:    Play step $h_\star + 1$ to $H$ using the UCBVI-UPLOW algorithm (Algorithm 4).
5: **end for**
6: Denote the final output of UCBVI-UPLOW as

$$(\overline{V}_{h_\star+1}, \underline{V}_{h_\star+1}, \widehat{\pi}^{\mathrm{UCB}}_{(h_\star+1):H}) \leftarrow \text{UCBVI-UPLOW}(n_{\mathrm{UCB}}).$$

7: # Stage 2: Learn step $1 : h_\star$ via executing $\mu$ + pessimistic offline policy optimization
8: Collect $\mathcal{D} \leftarrow \{n - n_{\mathrm{UCB}}$ episodes of data using policy $\mu$ up to step $h_\star\}$.
9: Learn policy $\widehat{\pi}^{\mathrm{PEVI}}_{1:h_\star}$ via the TRUNCATED-PEVI-ADV(Algorithm 5):

$$\widehat{\pi}^{\mathrm{PEVI}}_{1:h_\star} \leftarrow \text{TRUNCATED-PEVI-ADV}(\mathcal{D}, h_\star, \underline{V}_{h_\star+1}).$$

10: **return** Policy $\widehat{\pi} = (\widehat{\pi}^{\mathrm{PEVI}}_{1:h_\star}, \widehat{\pi}^{\mathrm{UCB}}_{(h_\star+1):H})$.

---

complexity $\widetilde{O}(H^3 S C^\star / \varepsilon^2)$. We investigate policy finetuning beyond the single-policy concentrability assumption in Section 5.

We also remark that Theorem 3 generalizes both the $\Omega(H^3 S A / \varepsilon^2)$ lower bound for online RL [11, 58, 13] into the policy finetuning problem, as well as the $\Omega(H^3 S C^\star / \varepsilon^2)$ lower bound for offline RL under single-policy concentrability with $C^\star \geq 2$ [42][5]. Further, Theorem 3 directly implies an $\Omega(H^3 S C^\star / \varepsilon^2)$ lower bound for offline RL with $2 \leq C^\star \leq O(A)$, as any algorithm for offline policy optimization is also an algorithm for policy finetuning via the offline reduction.

**Proof intuition; Construction of hard instance**   The proof of Theorem 3 constructs a family of hard MDPs that requires solving $HS$ "independent" bandit problems with $A$ arms, similar as in existing $\Omega(H^3 S A / \varepsilon^2)$ lower bounds for online RL [11, 58]. However, our key modification is that we let the optimal arms to be always within the first $K := \min\{C^\star, A\}$ actions instead of all $A$ actions, and we define our reference policy $\mu$ to play uniformly within $[K]$. This $\mu$ has the following properties:

- $\mu$ satisfies $C^\star$ single-policy concentrability for any MDP in this family (Lemma D.1).

- $\mu$ provides the knowledge that the optimal actions are within $[K]$, but *no other knowledge* about the optimal actions.

Therefore, with $\mu$ at hand, any policy finetuning algorithm can "gain the knowledge" that the optimal actions are within $[K]$, but still needs to try all $K$ actions in order to solve each bandit problem—rigorizing this information-theoretically gives the $\Omega(H^3 S K / \varepsilon^2) = \Omega(H^3 S \min\{C^\star, A\} / \varepsilon^2)$ lower bound.

## 5   Hybrid offline/online algorithm for policy finetuning

Towards circumventing the lower bound in Theorem 3, in this section, we study policy finetuning under more relaxed assumptions on the reference policy $\mu$. A weaker $\mu$ will induce a higher sample complexity for naive offline reduction approaches, and thus yields opportunities for designing new algorithms that can potentially better utilize $\mu$.

More concretely, we consider the following relaxation: We assume $\mu$ satisfies *partial concentrability* only up to a certain time-step $h_\star \leq H$, and may not have any bounded concentrability at steps $h > h_\star$. We formalize this in the following

---

[5]The lower bound in [42] is $\Omega(S C^\star / \varepsilon^2 (1 - \gamma)^3)$ for the infinite-horizon $\gamma$-discounted setting, which corresponds to an $\Omega(H^3 S C^\star / \varepsilon^2)$ lower bound for our finite-horizon setting.

**Assumption B** ($h_\star$-partial concentrability). *The reference policy $\mu$ satisfies the single-policy concentrability with respect to $\pi_\star$ up to step $h_\star$ only:*

$$\max_{h \leq h_\star} \max_{s,a \in \mathcal{S} \times \mathcal{A}} \frac{d_h^{\pi_\star}(s,a)}{d_h^\mu(s,a)} \leq C^{\text{partial}}$$

*(with the convention $0/0 = 0$), where $\pi_\star$ is some deterministic optimal policy of the MDP, and constant $C^{\text{partial}} \geq 1$.*

**Algorithm description**    We design a hybrid offline/online algorithm HOOVI (presented in Algorithm 2) for policy finetuning under the partial concentrability assumption. At a high-level, the algorithm consists of two main stages:

- In the first stage, it runs an online algorithm UCBVI-UPLOW which uses optimistic exploration to find a near-optimal policy $\widehat{\pi}^{\text{UCB}}$ and an accurate value estimate for steps $(h_\star + 1) : H$.
- In the second stage, we run a TRUNCATED-PEVI-ADV algorithm, which collects data from $\mu$ and runs offline policy optimization to find a near-optimal policy $\widehat{\pi}^{\text{PEVI}}$ for steps $1 : h_\star$, *building on the lower value estimate $\underline{V}_{h_\star+1}$ from the first stage.*

This strategy makes sense intuitively as the reference policy $\mu$ does not have guarantees for steps $h_\star + 1 : H$ and thus the algorithm is required to perform optimistic exploration first to get a good policy. However, additional technical cares are needed in order to make the above algorithm provably sample-efficient. The analysis of the second stage requires the online algorithm in the first stage to not only perform fast exploration (e.g. by using upper confidence bounds), but also output a *lower value estimate* for step $h_\star + 1$, and in addition output a final output policy that achieves at least the value of the lower value estimate *at every state $s \in \mathcal{S}$*. Such lower bounds are not directly available in standard online RL algorithms such as UCBVI [5].

We resolve this by designing the UCBVI-UPLOW algorithm (detailed description in Algorithm 4), which is a modification of the Nash-VI Algorithm of Liu et al. [34] (for two-player Markov games) into the single-player case. This algorithm is particularly suitable for our purpose since it maintains both upper bounds of $V^\star$ and lower bounds for the value function of the deployed policies. Our UCBVI-UPLOW further integrates the certified policy technique of Bai et al. [7] to make sure that its output policy achieves value greater or equal than the lower bound at every state (similar guarantees can also be obtained by the policy certificate technique of Dann et al. [12]).

We now state our main theoretical guarantee for the HOOVI algorithm. The proof can be found in Appendix E.

**Theorem 4** (Hybrid online / offline learning for policy finetuning). *Suppose the reference policy $\mu$ satisfies the partial concentrability (Assumption B) up to some step $h_\star \leq H$. Then for small enough $\varepsilon \leq \min\left\{ h_\star^{-2.5}, C^{\text{partial}}/S \right\}$, HOOVI (Algorithm 2) outputs a policy $\widehat{\pi}$ such that $V_1^\star(s_1) - V_1^{\widehat{\pi}}(s_1) \leq \varepsilon$ with probability at least $1 - \delta$, within*

$$n = \widetilde{O}\left( \frac{H^2 h_\star S C^{\text{partial}} + (H - h_\star)^3 SA (C^{\text{partial}})^2}{\varepsilon^2} \right)$$

*episodes of play.*

**Comparison against offline reduction and purely online algorithms**    The sample complexity in Theorem 4 compares favorably against both naive offline reduction as well as purely online algorithms in certain situations. First, naive offline reduction with $\mu$ does not have any guarantee since $\mu$ is not assumed to have a finite single-policy concentrability at $h \geq h_\star + 1$. We can modify $\mu$ into $\mu'$ that plays uniformly within $\mathcal{A}$ at steps $h \geq h_\star + 1$; the single-policy concentrability coefficient of $\mu'$ is guaranteed to be finite but scales exponentially as $O(A^{H-h_\star})$ in the worst case, leading to a sample complexity much worse than ours (which is polynomial in $H, S, A$).

On the other hand, a sharp online algorithm can still achieve $\widetilde{O}(H^3 SA/\varepsilon^2)$ in this setting (by optimistic exploration from scratch). Our Theorem 4 is in general incomparable with this, but can be better in cases when both $C^{\text{partial}}$ and $H - h_\star$ are small, e.g., if $C^{\text{partial}} = o(A)$ and $(H - h_\star)/H = o((C^{\text{partial}})^{-2/3})$. This makes sense intuitively as our hybrid offline/online algorithm benefits the most if the length requiring exploration $(H - h_\star)$ is small, and the partial concentrability $C^{\text{partial}}$ is small so that $\mu$ still has a high-quality for the first $h_\star$ steps. To best of our knowledge, this is first result that characterizes when the sample complexity of such hybrid algorithms can be beneficial over purely online or offline algorithms.

# 6 Conclusion & discussions

This paper studies policy finetuning, a new reinforcement learning setting that allows us to compare and connect sample-efficient online and offline reinforcement learning. We establish sharp upper and lower bounds for policy finetuning under various assumptions on the reference policy. Our bounds show that the optimal policy finetuning algorithm is either offline reduction or a purely online algorithm in the specific setting where the reference policy satisfies single-policy concentrability, and we also show that a hybrid online/offline algorithm can be advantageous over both in more relaxed settings. Many directions could be of interest for future research, such as alternative assumptions on the reference policy, or policy finetuning with function approximation.

Also, while our contributions are mainly theoretical, implementing or extending our policy finetuning algorithms on real-world RL tasks would be a compelling future direction. When the environment is a tabular MDP, our Algorithm 1 (offline reduction) and Algorithm 2 (hybrid offline / online RL) are readily implementable. When there is large state/action space and potentially function approximation, we believe our algorithm can be adapted, for example, by replacing all the optimistic/pessimistic value iteration steps by DQN-type algorithms [38] with positive/negative bonus functions [50]. Experimental evaluation of such algorithms would be a good direction for future work.

## Acknowledgment

The authors would like to thank Ming Yin, Chi Jin and David Forsyth for the many insightful discussions. NJ acknowledges funding support from the ARL Cooperative Agreement W911NF-17-2-0196, NSF IIS-2112471, and Adobe Data Science Research Award. HW, CX, YB are funded through employment with Salesforce.

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
