---

**Algorithm 3** Value Iteration with Lower Confidence Bounds (VI-LCB) for episodic MDPs

---

**Require:** Offline dataset $\mathcal{D} = \left\{ (s_1^{(i)}, a_1^{(i)}, r_1^{(i)}, \ldots, s_H^{(i)}, a_H^{(i)}, r_H^{(i)}) \right\}_{i=1}^n$.

1: Randomly split the dataset $\mathcal{D}$ into $\{\mathcal{D}_h\}_{h=1}^H$ with $|\mathcal{D}_h| = n/H$.
2: Let $N_h(s, a)$ and $N_h(s, a, s')$ denote the visitation count of $(s, a)$ and $(s, a, s')$ at step $h$ within dataset $\mathcal{D}_h$. Construct empirical model estimates:

$$\widehat{r}_h(s, a) \leftarrow r_h(s, a) \mathbb{1}\left\{ N_h(s, a) \geq 1 \right\},$$

$$\widehat{\mathbb{P}}_h(s'|s, a) \leftarrow \frac{N_h(s, a, s')}{N_h(s, a) \vee 1}.$$

3: Set $\widehat{V}_{H+1}(s) \leftarrow 0$ for all $s \in \mathcal{S}$.
4: **for** $h = H, \ldots, 1$ **do**
5:     Set $b_h(s, a) \leftarrow c \cdot \sqrt{\frac{H^2 \iota}{N_h(s,a) \vee 1}}$ (where $\iota := \log(HSA/\delta)$).
6:     Perform value update for all $(s, a)$:

$$\widehat{Q}_h(s, a) \leftarrow \widehat{r}_h(s, a) + \left[ \widehat{\mathbb{P}}_h \widehat{V}_{h+1} \right](s, a) - b_h(s, a);$$

$$\widehat{V}_h(s) \leftarrow \left[ \max_a \widehat{Q}_h(s, a) \right] \vee 0.$$

7:     Set $\widehat{\pi}_h(s) \leftarrow \arg\max_a \widehat{Q}_h(s, a)$ for all $s \in \mathcal{S}$.
8: **end for**
9: **return** Value estimate $\widehat{V} = \left\{ \widehat{V}_h \right\}_{h \in [H]}$, policy $\widehat{\pi} = \{\widehat{\pi}_h\}_{h \in [H]}$.

---

## A  Technical tools

**Lemma A.1** (Binomial concentration). *Suppose $N \sim \text{Bin}(n, p)$ where $n \geq 1$ and $p \in [0, 1]$. Then with probability at least $1 - \delta$, we have*

$$\frac{p}{N \vee 1} \leq \frac{8 \log(1/\delta)}{n}.$$

*Proof.* If $p \leq 8 \log(1/\delta)/n$, the result clearly holds regardless of the value of $N$. Suppose $p > 8 \log(1/\delta)/n$, then by the multiplicative Chernoff bound, we have

$$\mathbb{P}\left( N < \frac{1}{2}np \right) \leq \exp\left( -\frac{(1/2)^2 np}{2} \right) = \exp(-np/8) \leq \delta.$$

Therefore, with probability at least $1 - \delta$, we have $N \geq np/2$ and thus

$$\frac{p}{N \vee 1} \leq \frac{p}{N} \leq \frac{2}{n} \leq \frac{8 \log(1/\delta)}{n}.$$

This shows the desired result. $\qquad\qquad\qquad\qquad\qquad\qquad\qquad\qquad\qquad\qquad\qquad\qquad\qquad\square$

## B  Proof of Theorem 1

Throughout the proofs we let $C > 0$ denote an absolute constant that can vary from line to line. This $C$ is not to be confused with the single-policy concentrability coefficient $C^\star$ in Assumption A.

### B.1  Algorithm

We first present the VI-LCB algorithm in Algorithm 3. This algorithm is an analogue of the original VI-LCB algorithm of [42] for finite-horizon MDPs (instead of infinite-horizon discounted MDPs). Within the algorithm, the constant $c$ in Line 5 is chosen to be the same $c > 0$ as in Lemma B.1.

## B.2 Some lemmas

**Lemma B.1** (Concentration). *Under the setting of Theorem 1, there exists an absolute constant $c > 0$ such that the concentration event $\mathcal{E}$ holds with probability at least $1 - \delta$, where*

$$\mathcal{E} := \left\{ \left| [\widehat{r}_h - r_h](s, a) + \left[ (\widehat{\mathbb{P}}_h - \mathbb{P}_h)\widehat{V}_{h+1} \right](s, a) \right| \le c \cdot \sqrt{\frac{H^2 \iota}{N_h(s,a) \vee 1}} = b_h(s, a), \quad and \right.$$

$$\left. \frac{1}{N_h(s,a) \vee 1} \le c \cdot \frac{H\iota}{n d_h^\mu(s,a)} \quad for\ all\ (h, s, a) \in [H] \times \mathcal{S} \times \mathcal{A} \right\},$$

*and $\iota := \log(HSA/\delta)$.*

*Proof of Lemma B.1.* Fix any $(h, s, a)$. The first claim holds trivially if $N_h(s, a) = 0$, as $\widehat{r}_h(s, a) = \widehat{\mathbb{P}}_h(\cdot|s, a) = 0$ by definition, and thus the left-hand side is upper bounded by $H + 1 \le 2H \le 2\sqrt{H^2 \iota}$. If $N_h(s, a) \ge 1$, conditioned on $N_h(s, a)$, we have

$$\widehat{r}_h(s, a) = r_h(s, a),$$

$$\left| \left[ (\widehat{\mathbb{P}}_h - \mathbb{P}_h)\widehat{V}_{h+1} \right](s, a) \right| \le c \cdot \sqrt{\frac{H^2 \log(HSA/\delta)}{N_h(s, a)}},$$

where the last inequality is obtained by the Azuma-Hoeffding inequality with probability at least $1 - \delta/(2HSA)$, using the fact that the data used in obtaining $\widehat{\mathbb{P}}_h$ is independent of the data used in obtaining $\widehat{V}_{h+1}$ (due to the data splitting in Algorithm 3). Therefore

$$\left| [\widehat{r}_h - r_h](s, a) + \left[ (\widehat{\mathbb{P}}_h - \mathbb{P}_h)\widehat{V}_{h+1} \right](s, a) \right| \le c \cdot \sqrt{\frac{H^2 \iota}{N_h(s, a) \vee 1}}.$$

Further taking the union bound yields the first claim over all $(h, s, a)$ with probability at least $1 - \delta/2$.

For the second claim, notice that $N_h(s, a) \sim \mathsf{Bin}(n/H, d_h^\mu(s, a))$. Applying Lemma A.1 yields that

$$\frac{1}{N_h(s,a) \vee 1} \le \frac{8 \log(2HSA/\delta)}{n/H \cdot d_h^\mu(s, a)} \le c \cdot \frac{H\iota}{n d_h^\mu(s, a)}$$

with probability at least $1 - \delta/(2HSA)$. Taking the union bound yields the second claim over all $(h, s, a)$ with probability at least $1 - \delta/2$. $\square$

**Lemma B.2** (Monotonicity for VI-LCB). *Let $\widehat{\pi}$ be the output policy of Algorithm 3. Then, on the event $\mathcal{E}$ defined in Lemma B.1, we have*

$$\widehat{V}_h(s) \le V_h^{\widehat{\pi}}(s) \le V_h^\star(s)$$

*for any $s \in \mathcal{S}$ and $h \in [H]$.*

*Proof of Lemma B.2.* We first prove $\widehat{V}_h(s) \le V_h^{\widehat{\pi}}(s)$ for any $s \in \mathcal{S}$ and $h \in [H]$ by induction. For $h = H$, and any $s \in \mathcal{S}, a \in \mathcal{A}$,

$$Q_H^{\widehat{\pi}}(s, a) - \widehat{Q}_H(s, a)$$
$$= r_H(s, a) - \widehat{r}_H(s, a) + b_H(s, a) \ge 0,$$

where the last inequality is by Lemma B.1. Then,

$$V_H^{\widehat{\pi}}(s) - \widehat{V}_H(s)$$
$$= Q_H^{\widehat{\pi}}(s, \widehat{\pi}_H(s)) - \left[ \max_a \widehat{Q}_H(s, a) \right] \vee 0$$
$$= Q_H^{\widehat{\pi}}(s, a_{s,H}) - \left[ \widehat{Q}_H(s, a_{s,H}) \right] \vee 0 \qquad (a_{s,H} := \widehat{\pi}_H(s))$$
$$= \left[ Q_H^{\widehat{\pi}}(s, a_{s,H}) - \widehat{Q}_H(s, a_{s,H}) \right] \wedge Q_H^{\widehat{\pi}}(s, a_{s,H}) \ge 0,$$

where the last inequality follows from Lemma B.1, and $Q_H^\pi(s,a) \in [0,1]$ for any $\pi$, $s \in \mathcal{S}$, and $a \in \mathcal{A}$.

We now show that, if $\widehat{V}_{h+1}(s) \le V_{h+1}^{\widehat{\pi}}(s)$ holds for any $s$, we also have $\widehat{V}_h(s) \le V_h^{\widehat{\pi}}(s)$ for any $s$. Recall that $\widehat{V}_h(s) = \max_a \widehat{Q}_h(s,a) \vee 0 = \widehat{Q}_h(s, \widehat{\pi}_h(s)) \vee 0$. The claim clearly holds in the trivial case of $\widehat{V}_h(s) = 0$. Otherwise, we have

$$
\begin{aligned}
&V_h^{\widehat{\pi}}(s) - \widehat{V}_h(s) \\
&= Q_h^{\widehat{\pi}}(s, a_{s,h}) - \widehat{Q}_h(s, a_{s,h}) && (a_{s,h} := \widehat{\pi}_h(s)) \\
&= r_h(s, a_{s,h}) + \left[\mathbb{P}_h V_{h+1}^{\widehat{\pi}}\right](s, a_{s,h}) - \widehat{r}_h(s, a_{s,h}) - \left[\widehat{\mathbb{P}}_h \widehat{V}_{h+1}\right](s, a_{s,h}) + b_h(s, a_{s,h}) \\
&= r_h(s, a_{s,h}) - \widehat{r}_h(s, a_{s,h}) + \left[\mathbb{P}_h\left(V_{h+1}^{\widehat{\pi}} - \widehat{V}_{h+1}\right)\right](s, a_{s,h}) + \left[\left(\mathbb{P}_h - \widehat{\mathbb{P}}_h\right)\widehat{V}_{h+1}\right](s, a_{s,h}) + b_h(s, a_{s,h}) \\
&\ge r_h(s, a_{s,h}) - \widehat{r}_h(s, a_{s,h}) + \left[\left(\mathbb{P}_h - \widehat{\mathbb{P}}_h\right)\widehat{V}_{h+1}\right](s, a_{s,h}) + b_h(s, a_{s,h}) \\
&\ge 0,
\end{aligned}
$$

where the first inequality follows from $V_{h+1}^{\widehat{\pi}}(s) \ge \widehat{V}_{h+1}(s)$ for any $s$, and the last inequality is by Lemma B.1. This completes the proof of $\widehat{V}_h(s) \le V_h^{\widehat{\pi}}(s)$, $\forall s \in \mathcal{S}, h \in [H]$.

The argument $V_h^{\widehat{\pi}}(s) \le V_h^\star(s)$ holds by definition of $V_h^\star(s) = V_h^{\pi^\star}(s) \ge V_h^\pi(s)$, for any $\pi$, $s \in \mathcal{S}$, and $h \in [H]$. Thus, we complete the proof. $\qquad\square$

**Lemma B.3** (Performance decomposition for VI-LCB). *On the event $\mathcal{E}$ defined in Lemma B.1, we have*

$$
\sum_{s \in \mathcal{S}} d_h^{\pi^\star}(s)(V_h^\star(s) - \widehat{V}_h(s)) \le 2 \sum_{h'=h}^{H} \sum_{(s,a) \in \mathcal{S} \times \mathcal{A}} d_{h'}^{\pi^\star}(s, a) b_{h'}(s, a),
$$

*for any $h \in [H]$.*

*Proof of Lemma B.3.* Throughout this proof we let $\widehat{Q}_h(s, \pi_\star) := \widehat{Q}_h(s, \pi_{\star,h}(s))$ for shorthand. We have

$$
\begin{aligned}
&\sum_{s \in \mathcal{S}} d_h^{\pi^\star}(s)(V_h^\star(s) - \widehat{V}_h(s)) \\
&\le \sum_{s \in \mathcal{S}} d_h^{\pi^\star}(s)(V_h^\star(s) - \max_a \widehat{Q}_h(s,a)) && \text{(by definition of } \widehat{V}_h(s) = \max_a \widehat{Q}_h(s,a) \vee 0) \\
&\le \sum_{s \in \mathcal{S}} d_h^{\pi^\star}(s)(V_h^\star(s) - \widehat{Q}_h(s, \pi_\star)) \\
&= \sum_{h'=h}^{H} \sum_{(s,a) \in \mathcal{S} \times \mathcal{A}} d_{h'}^{\pi^\star}(s,a) r_{h'}(s,a) - \sum_{s \in \mathcal{S}} d_h^{\pi^\star}(s) \widehat{Q}_h(s_1, \pi_\star) \\
&= \sum_{h'=h}^{H} \sum_{(s,a) \in \mathcal{S} \times \mathcal{A}} d_{h'}^{\pi^\star}(s,a) r_{h'}(s,a) - \sum_{h'=h}^{H} \sum_{s,a,s',a'} \left(d_{h'}^{\pi^\star}(s,a) \widehat{Q}_{h'}(s,a) - d_{h'+1}^{\pi^\star}(s',a') \widehat{Q}_{h'+1}(s',a')\right) \\
&\le \sum_{h'=h}^{H} \sum_{(s,a) \in \mathcal{S} \times \mathcal{A}} d_{h'}^{\pi^\star}(s,a) r_{h'}(s,a) - \sum_{h'=h}^{H} \sum_{s,a,s'} \left(d_{h'}^{\pi^\star}(s,a) \widehat{Q}_{h'}(s,a) - d_{h'+1}^{\pi^\star}(s') \widehat{V}_{h'+1}(s')\right) \\
&\qquad\qquad\qquad\qquad\qquad\qquad\qquad\qquad\qquad\qquad\qquad\qquad\qquad\qquad \text{(by definition of } \widehat{V}_{h'}) \\
&= \sum_{h'=h}^{H} \sum_{(s,a) \in \mathcal{S} \times \mathcal{A}} d_{h'}^{\pi^\star}(s,a) r_h(s,a) - \sum_{h'=h}^{H} \sum_{(s,a) \in \mathcal{S} \times \mathcal{A}} d_{h'}^{\pi^\star}(s,a) \left(\widehat{Q}_{h'}(s,a) - \left[\mathbb{P}_{h'} \widehat{V}_{h'+1}\right](s,a)\right)
\end{aligned}
$$

$$= \sum_{h'=h}^{H} \sum_{(s,a)\in\mathcal{S}\times\mathcal{A}} d_{h'}^{\pi_\star}(s,a) \left( r_{h'}(s,a) + \left[ \mathbb{P}_{h'} \widehat{V}_{h'+1} \right](s,a) - \widehat{Q}_{h'}(s,a) \right)$$

$$= \sum_{h'=h}^{H} \sum_{(s,a)\in\mathcal{S}\times\mathcal{A}} d_{h'}^{\pi_\star}(s,a) \left( r_{h'}(s,a) + \left[ \mathbb{P}_{h'} \widehat{V}_{h'+1} \right](s,a) - \widehat{r}_{h'}(s,a) - \left[ \widehat{\mathbb{P}}_{h'} \widehat{V}_{h'+1} \right](s,a) + b_{h'}(s,a) \right)$$

$$\leq 2 \sum_{h'=h}^{H} \sum_{(s,a)\in\mathcal{S}\times\mathcal{A}} d_{h'}^{\pi_\star}(s,a) b_{h'}(s,a). \qquad \text{(by the concentration event } \mathcal{E})$$

This completes the proof. $\qquad\qquad\square$

### B.3    Proof of main theorem

We are now ready to prove the Theroem 1. We first prove part (a). By Lemma B.2, we have

$$\max_{h\in[H]} \sum_{s\in\mathcal{S}} d_h^{\pi_\star}(s)(V_h^\star(s) - \widehat{V}_h(s))$$

$$\leq 2 \sum_{h=1}^{H} \sum_{(s,a)\in\mathcal{S}\times\mathcal{A}} d_h^{\pi_\star}(s,a) b_h(s,a)$$

$$= c \sum_{h=1}^{H} \sum_{(s,a)\in\mathcal{S}\times\mathcal{A}} d_h^{\pi_\star}(s,a) \cdot \sqrt{\frac{H^2 \iota}{N_h(s,a) \vee 1}}$$

$$= c\sqrt{H^2 \iota} \sum_{h=1}^{H} \sum_{(s,a)\in\mathcal{S}\times\mathcal{A}} d_h^{\pi_\star}(s,a) \cdot \sqrt{\frac{1}{N_h(s,a) \vee 1}}$$

$$\leq c\sqrt{H^2 \iota} \sum_{h=1}^{H} \sum_{(s,a)\in\mathcal{S}\times\mathcal{A}} d_h^{\pi_\star}(s,a) \cdot \sqrt{\frac{H \iota}{n d_h^\mu(s,a)}} \qquad \text{(by the concentration event } \mathcal{E})$$

$$\leq c\sqrt{H^3 \iota^2} \sum_{h=1}^{H} \sum_{(s,a)\in\mathcal{S}\times\mathcal{A}} \sqrt{d_h^{\pi_\star}(s,a)} \cdot \sqrt{\frac{d_h^{\pi_\star}(s,a)}{n d_h^\mu(s,a)}}$$

$$\leq c\sqrt{\frac{H^3 C^\star \iota^2}{n}} \sum_{h=1}^{H} \sum_{(s,a)\in\mathcal{S}\times\mathcal{A}} \sqrt{d_h^{\pi_\star}(s,a)} \qquad \text{(Assumption A)}$$

$$= c\sqrt{\frac{H^3 C^\star \iota^2}{n}} \sum_{h=1}^{H} \sum_{(s,a)\in\mathcal{S}\times\mathcal{A}} \sqrt{\mathbb{1}\{a = \pi_\star(s)\} \cdot d_h^{\pi_\star}(s,a)} \qquad (\pi_\star \text{ is deterministic})$$

$$\leq c\sqrt{\frac{H^3 C^\star \iota^2}{n}} \sqrt{\sum_{h=1}^{H} \sum_{(s,a)\in\mathcal{S}\times\mathcal{A}} \mathbb{1}\{a = \pi_\star(s)\}} \cdot \sqrt{\sum_{h=1}^{H} \sum_{(s,a)\in\mathcal{S}\times\mathcal{A}} d_h^\star(s,a)}$$

$$\text{(by Cauchy–Schwarz inequality)}$$

$$\leq c\sqrt{\frac{H^3 C^\star \iota^2}{n}} \sqrt{HS} \cdot \sqrt{H}$$

$$= c\sqrt{\frac{H^5 S C^\star \iota^2}{n}}.$$

Therefore, as long as $n \geq O(H^5 S C^\star \iota^2 / \varepsilon^2)$, we have $\max_{h\in[H]} \sum_{s\in\mathcal{S}} d_h^{\pi_\star}(s)(V_h^\star(s) - \widehat{V}_h(s)) \leq \varepsilon$. This shows part (a). Further, this bound at $h = 1$ says

$$V_1^\star(s_1) - \widehat{V}_1(s_1) \leq \varepsilon$$

(as we assumed deterministic $s_1$). Part (b) follows directly from this and the fact that $V_1^{\widehat{\pi}}(s_1) \geq \widehat{V}_1(s_1)$ which was shown in Lemma B.2. $\qquad\square$

# C  Proof of Theorem 2

## C.1  Some Lemmas

**Lemma C.1** (Concentration). *Under the setting of Theorem 2, there exists an absolute constant $c > 0$ such that the concentration event $\mathcal{E}$ holds with probability at least $1 - \delta$, where*

$$\mathcal{E} := \left\{ (i): \left| [\widehat{r}_{h,0} - r_h](s,a) + \left[ (\widehat{\mathbb{P}}_{h,0} - \mathbb{P}_h)\widehat{V}_{h+1}^{\mathrm{ref}} \right](s,a) \right| \right.$$

$$\leq c \cdot \left( \sqrt{\frac{[\widehat{\mathbb{V}}_{h,0}(\widehat{V}_{h+1}^{\mathrm{ref}})](s,a)\iota}{N_{h,0}(s,a) \vee 1}} + \frac{H\iota}{N_{h,0}(s,a) \vee 1} \right) = b_{h,0}(s,a),$$

$$(ii): \left| \left[ (\widehat{\mathbb{P}}_{h,1} - \mathbb{P}_h)(\widehat{V}_{h+1} - \widehat{V}_{h+1}^{\mathrm{ref}}) \right](s,a) \right|$$

$$\leq c \cdot \left( \sqrt{\frac{[\widehat{\mathbb{V}}_{h,1}(\widehat{V}_{h+1} - \widehat{V}_{h+1}^{\mathrm{ref}})](s,a)\iota}{N_{h,1}(s,a) \vee 1}} + \frac{H\iota}{N_{h,1}(s,a) \vee 1} \right) = b_{h,1}(s,a),$$

$$(iii): \frac{1}{N_{h,0}(s,a) \vee 1} \leq c \cdot \frac{\iota}{n d_h^\mu(s,a)} \quad and$$

$$\left. (iv): \frac{1}{N_{h,1}(s,a) \vee 1} \leq c \cdot \frac{H\iota}{n d_h^\mu(s,a)} \quad for\ all\ (h,s,a) \in [H] \times \mathcal{S} \times \mathcal{A} \right\},$$

*and $\iota := \log(HSA/\delta)$.*

*Proof of Lemma C.1.* Claim (i) holds trivially if $N_{h,0}(s,a) = 0$, as $\widehat{r}_{h,0}(s,a) = \widehat{\mathbb{P}}_{h,0}(\cdot|s,a) = 0$ by definition, and thus the left-hand side is upper bounded by $H + 1 \leq 2H \leq H\iota$. If $N_{h,0}(s,a) \geq 1$, conditioned on $N_{h,0}(s,a)$, we have

$$\widehat{r}_{h,0}(s,a) = r_h(s,a),$$

$$\left| \left[ (\widehat{\mathbb{P}}_{h,0} - \mathbb{P}_h)\widehat{V}_{h+1}^{\mathrm{ref}} \right](s,a) \right| \leq c \cdot \left( \sqrt{\frac{[\widehat{\mathbb{V}}_{h,0}(\widehat{V}_{h+1}^{\mathrm{ref}})](s,a) \log(HSA/\delta)}{N_{h,0}(s,a) \vee 1}} + \frac{H \log(HSA/\delta)}{N_{h,0}(s,a) \vee 1} \right)$$

where the last inequality is obtained by the empirical Bernstein inequality [37, Theorem 4] with probability at least $1 - \delta/(2HSA)$. Therefore

$$\left| [\widehat{r}_{h,0} - r_h](s,a) + \left[ (\widehat{\mathbb{P}}_{h,0} - \mathbb{P}_h)\widehat{V}_{h+1}^{\mathrm{ref}} \right](s,a) \right| \leq c \cdot \left( \sqrt{\frac{[\widehat{\mathbb{V}}_{h,0}(\widehat{V}_{h+1}^{\mathrm{ref}})](s,a)\iota}{N_{h,0}(s,a) \vee 1}} + \frac{H\iota}{N_{h,0}(s,a) \vee 1} \right).$$

Further taking the union bound yields the first claim over all $(h,s,a)$ with probability at least $1 - \delta/4$. The claim (ii) also follows from the similar argument. Claims (iii) and (iv) can be obtained directly from Lemma A.1 and a union bound, in a similar fashion as in the proof of Lemma B.1. (Note that $N_{h,0}(s,a) \sim \mathsf{Bin}(n_0, d_h^\mu(s,a))$ and $N_{h,1} \sim \mathsf{Bin}(n_{1,h}, d_h^\mu(s,a))$ where $n_0 = n/3$ and $n_{1,h} = n/(3H)$ due to our data splitting schedule.) This completes the proof. $\qquad \square$

**Lemma C.2** (Monotonicity for PEVI-ADV). *Let $\widehat{\pi}$ be the output policy of Algorithm 1. Then, on the event $\mathcal{E}$ defined in Lemma C.1, we have*

$$\widehat{V}_h(s) \leq V_h^{\widehat{\pi}}(s) \leq V_h^\star(s)$$

*for any $s \in \mathcal{S}$ and $h \in [H]$.*

*Proof of Lemma C.2.* We provide the proof by induction. For $h = H$, and any $s, a$,

$$\begin{aligned} Q_H^{\widehat{\pi}}(s,a) &- \widehat{Q}_H(s,a) \\ &= r_H(s,a) - \widehat{r}_{H,0}(s,a) + b_{H,0}(s,a). \end{aligned} \tag{2}$$

Eq.(2) is non-negative the concentration event $\mathcal{E}$(i) by Lemma C.1. Thus, we have $Q_H^{\widehat{\pi}}(s,a) - \widehat{Q}_H(s,a) \geq 0$ for any $s, a$, and then

$$
\begin{aligned}
&V_H^{\widehat{\pi}}(s) - \widehat{V}_H(s) \\
&= Q_H^{\widehat{\pi}}(s, \widehat{\pi}_H(s)) - \left[\max_a \widehat{Q}_H(s,a)\right] \vee 0 \\
&= Q_H^{\widehat{\pi}}(s, a_{s,H}) - \left[\widehat{Q}_H(s, a_{s,H})\right] \vee 0 && (a_{s,H} := \widehat{\pi}_H(s)) \\
&= \left[Q_H^{\widehat{\pi}}(s, a_{s,H}) - \widehat{Q}_H(s, a_{s,H})\right] \wedge Q_H^{\widehat{\pi}}(s, a_{s,H}) \\
&\geq 0,
\end{aligned}
$$

where the last inequality follows from the result of Eq.(2) is positive, and $Q_H^\pi(s,a) \in [0,1]$ for any $\pi, s \in \mathcal{S}, a \in \mathcal{A}$.

We now show that, if $\widehat{V}_{h+1}(s) \leq V_{h+1}^{\widehat{\pi}}(s)$ holds for any $s$, we also have $\widehat{V}_h(s) \leq V_h^{\widehat{\pi}}(s)$ for any $s$. Recall that $\widehat{V}_h(s) = \max_a \widehat{Q}_h(s,a) \vee 0 = \widehat{Q}_h(s, \widehat{\pi}_h(s)) \vee 0$. The claim clearly holds in the trivial case of $\widehat{V}_h(s) = 0$. Otherwise, we have

$$
\begin{aligned}
&V_h^{\widehat{\pi}}(s) - \widehat{V}_h(s) \\
&= Q_h^{\widehat{\pi}}(s, a_{s,h}) - \widehat{Q}_h(s, a_{s,h}) && (a_{s,h} := \widehat{\pi}_h(s)) \\
&= r_h(s, a_{s,h}) + \left[\mathbb{P}_h V_{h+1}^{\widehat{\pi}}\right](s, a_{s,h}) - \widehat{r}_{h,0}(s, a_{s,h}) - \left[\widehat{\mathbb{P}}_{h,0} \widehat{V}_{h+1}^{\mathrm{ref}}\right](s, a_{s,h}) \\
&\quad - \left[\widehat{\mathbb{P}}_h (\widehat{V}_{h+1} - \widehat{V}_{h+1}^{\mathrm{ref}})\right](s, a_{s,h}) + b_{h,0}(s, a_{s,h}) + b_{h,1}(s, a_{s,h}) \\
&= r_h(s, a_{s,h}) - \widehat{r}_{h,0}(s, a_{s,h}) + \left[\mathbb{P}_h \left(V_{h+1}^{\widehat{\pi}} - \widehat{V}_{h+1}\right)\right](s, a_{s,h}) + \left[\left(\mathbb{P}_h - \widehat{\mathbb{P}}_{h,0}\right) \widehat{V}_{h+1}^{\mathrm{ref}}\right](s, a_{s,h}) \\
&\quad + \left[\left(\mathbb{P}_h - \widehat{\mathbb{P}}_h\right)\left(\widehat{V}_{h+1} - \widehat{V}_{h+1}^{\mathrm{ref}}\right)\right](s, a_{s,h}) + b_{h,0}(s, a_{s,h}) + b_{h,1}(s, a_{s,h}) \\
&\geq r_h(s, a_{s,h}) - \widehat{r}_{h,0}(s, a_{s,h}) + \left[\left(\mathbb{P}_h - \widehat{\mathbb{P}}_{h,0}\right) \widehat{V}_{h+1}^{\mathrm{ref}}\right](s, a_{s,h}) && (3) \\
&\quad + \left[\left(\mathbb{P}_h - \widehat{\mathbb{P}}_h\right)\left(\widehat{V}_{h+1} - \widehat{V}_{h+1}^{\mathrm{ref}}\right)\right](s, a_{s,h}) + b_{h,0}(s, a_{s,h}) + b_{h,1}(s, a_{s,h})
\end{aligned}
$$

where the last inequality follows from $V_{h+1}^{\widehat{\pi}}(s) \geq \widehat{V}_{h+1}(s)$ for any $s$. Note that Eq.(3) is also non-negative under $\mathcal{E}$(i & ii) by Lemma C.1. This completes the proof of $\widehat{V}_h(s) \leq V_h^{\widehat{\pi}}(s)$, $\forall s \in \mathcal{S}, h \in [H]$. $\qquad \square$

**Lemma C.3** (Performance decomposition for PEVI-ADV). *On the event $\mathcal{E}$ defined in Lemma C.1, we have*

$$
\sum_{s \in \mathcal{S}} d_h^{\pi^\star}(s)(V_h^\star(s) - \widehat{V}_h(s)) \leq 2 \sum_{h'=h}^{H} \sum_{(s,a) \in \mathcal{S} \times \mathcal{A}} d_{h'}^{\pi^\star}(s,a)(b_{h',0}(s,a) + b_{h',1}(s,a)),
$$

*for any $h \in [H]$.*

*Proof of Lemma C.3.* Throughout this proof we let $\widehat{Q}_h(s, \pi_\star) := \widehat{Q}_h(s, \pi_{\star,h}(s))$ for shorthand. On the event $\mathcal{E}$, we have

$$
\begin{aligned}
&\sum_{s \in \mathcal{S}} d_h^{\pi^\star}(s)(V_h^\star(s) - \widehat{V}_h(s)) \\
&\leq \sum_{s \in \mathcal{S}} d_h^{\pi^\star}(s)(V_h^\star(s) - \max_a \widehat{Q}_h(s,a)) && \text{(by definition of } \widehat{V}_h(s) = \max_a \widehat{Q}_h(s,a) \vee 0) \\
&\leq \sum_{s \in \mathcal{S}} d_h^{\pi^\star}(s)(V_h^\star(s) - \widehat{Q}_h(s, \pi_\star)) \\
&= \sum_{h'=h}^{H} \sum_{(s,a) \in \mathcal{S} \times \mathcal{A}} d_{h'}^{\pi^\star}(s,a) r_{h'}(s,a) - \sum_{s \in \mathcal{S}} d_h^{\pi^\star}(s) \widehat{Q}_h(s_1, \pi_\star)
\end{aligned}
$$

$$= \sum_{h'=h}^{H} \sum_{(s,a)\in\mathcal{S}\times\mathcal{A}} d_{h'}^{\pi^\star}(s,a)r_{h'}(s,a) - \sum_{h'=h}^{H} \sum_{s,a,s',a'} \left( d_{h'}^{\pi^\star}(s,a)\widehat{Q}_{h'}(s,a) - d_{h'+1}^{\pi^\star}(s',a')\widehat{Q}_{h'+1}(s',a') \right)$$

$$\leq \sum_{h'=h}^{H} \sum_{(s,a)\in\mathcal{S}\times\mathcal{A}} d_{h'}^{\pi^\star}(s,a)r_{h'}(s,a) - \sum_{h'=h}^{H} \sum_{s,a,s'} \left( d_{h'}^{\pi^\star}(s,a)\widehat{Q}_{h'}(s,a) - d_{h'+1}^{\pi^\star}(s')\widehat{V}_{h'+1}(s') \right)$$

$$\text{(by definition of } \widehat{V}_{h'})$$

$$= \sum_{h'=h}^{H} \sum_{(s,a)\in\mathcal{S}\times\mathcal{A}} d_{h'}^{\pi^\star}(s,a)r_h(s,a) - \sum_{h'=h}^{H} \sum_{(s,a)\in\mathcal{S}\times\mathcal{A}} d_{h'}^{\pi^\star}(s,a)\left( \widehat{Q}_{h'}(s,a) - \left[ \mathbb{P}_{h'}\widehat{V}_{h'+1} \right](s,a) \right)$$

$$= \sum_{h'=h}^{H} \sum_{(s,a)\in\mathcal{S}\times\mathcal{A}} d_{h'}^{\pi^\star}(s,a)\left( r_{h'}(s,a) + \left[ \mathbb{P}_{h'}\widehat{V}_{h'+1} \right](s,a) - \widehat{Q}_{h'}(s,a) \right)$$

$$= \sum_{h'=h}^{H} \sum_{(s,a)\in\mathcal{S}\times\mathcal{A}} d_{h'}^{\pi^\star}(s,a)\Bigg( r_{h'}(s,a) - \widehat{r}_{h',0}(s,a) + \left[ \mathbb{P}_{h'}\widehat{V}_{h'+1} \right](s,a) - \left[ \widehat{\mathbb{P}}_{h',0}\widehat{V}_{h'+1}^{\mathrm{ref}} \right](s,a)$$

$$- \left[ \widehat{\mathbb{P}}_{h',1}(\widehat{V}_{h'+1} - \widehat{V}_{h'+1}^{\mathrm{ref}}) \right](s,a) + b_{h',0}(s,a) + b_{h',1}(s,a) \Bigg)$$

$$\leq 2 \sum_{h'=h}^{H} \sum_{(s,a)\in\mathcal{S}\times\mathcal{A}} d_{h'}^{\pi^\star}(s,a)(b_{h',0}(s,a) + b_{h',1}(s,a)). \qquad \text{(by Lemma C.1)}$$

This completes the proof. $\qquad\square$

**Lemma C.4** (Total variance lemma). *On the event $\mathcal{E}$ defined in Lemma C.1, the reference value function $\widehat{V}^{\mathrm{ref}}$ obtained in Algorithm 1 satisfies*

$$\sum_{h=1}^{H} \sum_{(s,a)\in\mathcal{S}\times\mathcal{A}} d_h^{\pi^\star}(s,a)\left[ \mathbb{V}_h\widehat{V}_{h+1}^{\mathrm{ref}} \right](s,a) \leq H^2 + c\sqrt{\frac{H^9 SC^\star \iota^2}{n_{\mathrm{ref}}}},$$

*where $c$ is an absolute constant.*

*Proof of Lemma C.4.* We first decompose our target as follows,

$$\sum_{h=1}^{H} \sum_{(s,a)\in\mathcal{S}\times\mathcal{A}} d_h^{\pi^\star}(s,a)[\mathbb{V}_h(\widehat{V}_{h+1}^{\mathrm{ref}})](s,a)$$

$$= \underbrace{\sum_{h=1}^{H} \sum_{(s,a)\in\mathcal{S}\times\mathcal{A}} d_h^{\pi^\star}(s,a)[\mathbb{V}_h V_{h+1}^\star](s,a)}_{\text{(I)}} + \underbrace{\sum_{h=1}^{H} \sum_{(s,a)\in\mathcal{S}\times\mathcal{A}} d_h^{\pi^\star}(s,a)[\mathbb{V}_h\widehat{V}_{h+1}^{\mathrm{ref}} - \mathbb{V}_h V_{h+1}^\star](s,a)}_{\text{(II)}}.$$

We now bound term (I) and term (II) separately.

Let $\mathcal{F}_{h+1}$ denote the $\sigma$-algebra that contains all information about the trajectory up to $s_{h+1}$ but not $a_{h+1}$. We have

$$\text{(I)} = \sum_{h=1}^{H} \mathbb{E}_{d^{\pi^\star}}\left[ \mathrm{Var}\left[ V_{h+1}^\star(s_{h+1}) \big| s_h, a_h \right] \right]$$

$$= \sum_{h=1}^{H} \mathbb{E}_{d^{\pi^\star}}\left[ \mathbb{E}\left[ \left( V_{h+1}^\star(s_{h+1}) + r_h(s_h, a_h) - V_h^\star(s_h) \right)^2 \big| s_h, a_h \right] \right]$$

$$\text{(by } \mathbb{E}[V_{h+1}^\star(s_{h+1})|s_h, a_h] = V_h^\star(s_h) - r_h(s_h, a_h))$$

$$= \sum_{h=1}^{H} \mathbb{E}_{d^{\pi^\star}}\left[ \left( V_{h+1}^\star(s_{h+1}) + r_h(s_h, a_h) - V_h^\star(s_h) \right)^2 \right]$$

$$= \sum_{h=1}^{H} \mathbb{E}_{d^{\pi_\star}} \left[ \left( V_{h+1}^\star(s_{h+1}) + r_h(s_h, a_h) - V_h^\star(s_h) \right)^2 \right]$$

$$+ 2 \underbrace{\sum_{1 \leq h < h' \leq H} \mathbb{E}_{d^{\pi_\star}} \left[ \left( V_{h+1}^\star(s_{h+1}) + r_h(s_h, a_h) - V_h^\star(s_h) \right) \cdot \left( V_{h'+1}^\star(s_{h'+1}) + r(s_h', a_h') - V_h^\star(s_h') \right) \right]}_{=0, \text{ because } \left( V_{h+1}^\star(s_{h+1}) + r_h(s_h, a_h) - V_h^\star(s_h) \right) \mathbb{E}_{d^{\pi_\star}} [V_{h'+1}^\star(s_{h'+1}) - V_{h'}^\star(s_{h'}) + r_{h'}(s_{h'}, a_{h'}) | \mathcal{F}_{h+1}] = 0 \text{ for any } h < h'}$$

$$= \mathbb{E}_{d^{\pi_\star}} \left[ \left( \sum_{h=1}^{H} \left( V_{h+1}^\star(s_{h+1}) + r_h(s_h, a_h) - V_h^\star(s_h) \right) \right)^2 \right]$$

$$= \mathbb{E}_{d^{\pi_\star}} \left[ \left( \sum_{h=1}^{H} r_h(s_h, a_h) + \sum_{h=1}^{H} \left( V_{h+1}^\star(s_{h+1}) - V_h^\star(s_h) \right) \right)^2 \right]$$

$$= \mathbb{E}_{d^{\pi_\star}} \left[ \left( \sum_{h=1}^{H} r_h(s_h, a_h) - V_1^\star(s_1) \right)^2 \right]$$

$$= \text{Var}_{d^{\pi_\star}} \left( \sum_{h=1}^{H} r_h(s_h, a_h) \right) \leq H^2.$$

For (II),

$$\text{(II)} = \sum_{h=1}^{H} \sum_{(s,a) \in \mathcal{S} \times \mathcal{A}} d_h^{\pi_\star}(s,a) [\mathbb{V}_h \widehat{V}_{h+1}^{\text{ref}} - \mathbb{V}_h V_{h+1}^\star](s,a)$$

$$= \sum_{h=1}^{H} \sum_{(s,a) \in \mathcal{S} \times \mathcal{A}} d_h^{\pi_\star}(s,a) \left[ \mathbb{P}_h \left( \widehat{V}_{h+1}^{\text{ref}} \right)^2 - \left( \mathbb{P}_h \widehat{V}_{h+1}^{\text{ref}} \right)^2 - \mathbb{P}_h \left( V_{h+1}^\star \right)^2 + \left( \mathbb{P}_h V_{h+1}^\star \right)^2 \right] (s,a)$$

$$\leq \sum_{h=1}^{H} \sum_{(s,a) \in \mathcal{S} \times \mathcal{A}} d_h^{\pi_\star}(s,a) \left[ \left| \mathbb{P}_h(\widehat{V}_{h+1}^{\text{ref}} + V_{h+1}^\star)(\widehat{V}_{h+1}^{\text{ref}} - V_{h+1}^\star) \right| \right.$$

$$\left. + \left| \mathbb{P}_h(\widehat{V}_{h+1}^{\text{ref}} + V_{h+1}^\star)\mathbb{P}_h(\widehat{V}_{h+1}^{\text{ref}} - V_{h+1}^\star) \right| \right] (s,a)$$

$$\leq 4H \sum_{h=1}^{H} \sum_{(s,a) \in \mathcal{S} \times \mathcal{A}} d_h^{\pi_\star}(s,a) \left[ \left| \mathbb{P}_h(\widehat{V}_{h+1}^{\text{ref}} - V_{h+1}^\star) \right| \right] (s,a)$$

$$\leq 4H \sum_{h=1}^{H} \sum_{(s,a) \in \mathcal{S} \times \mathcal{A}} d_{h+1}^{\pi_\star}(s') \left[ V_{h+1}^\star - \widehat{V}_{h+1}^{\text{ref}} \right] (s') \qquad \text{(As } \widehat{V}_{h+1}^{\text{ref}} \leq V_{h+1}^\star)$$

$$\leq 4H^2 \max_{h \in [H]} \sum_{(s,a) \in \mathcal{S} \times \mathcal{A}} d_h^{\pi_\star}(s) \left( V_h^\star - \widehat{V}_h^{\text{ref}} \right) (s)$$

$$\leq c \sqrt{\frac{H^9 S C^\star \iota^2}{n_{\text{ref}}}},$$

where the last inequality follows from Lemma C.5. Combining (I) and (II), we complete the proof. $\quad\square$

**Lemma C.5** (Guarantees for $\widehat{V}^{\text{ref}}$). *On the event $\mathcal{E}$ defined in Lemma C.1, the reference value function $\widehat{V}^{\text{ref}}$ obtained in Algorithm 1 satisfies*

$$\max_{h \in [H]} \sum_{s \in \mathcal{S}} d_h^{\pi_\star}(s) \left( V_h^\star(s) - \widehat{V}_h^{\text{ref}}(s) \right) \leq c \sqrt{\frac{H^5 S C^\star \iota^2}{n_{\text{ref}}}},$$

*where $c > 0$ is an absolute constant.*

*Proof of Lemma C.5.* This is a direct corollary of Theorem 1 (cf. the end of its proof in Section B.3).
□

## C.2 Proof of the Main Theorem

We now provide the proof of Theorem 2. We assume we are on the good event $\mathcal{E}$ defined in Lemma C.1, which happens with probability at least $1 - \delta$. By Lemma C.3, we know

$$\sum_{s \in \mathcal{S}} d_h^{\pi^\star}(s)(V_h^\star(s) - \widehat{V}_h(s)) \tag{4}$$

$$\leq 2 \underbrace{\sum_{h=1}^H \sum_{(s,a) \in \mathcal{S} \times \mathcal{A}} d_h^{\pi^\star}(s,a) b_{h,0}(s,a)}_{(I)} + 2 \underbrace{\sum_{h=1}^H \sum_{(s,a) \in \mathcal{S} \times \mathcal{A}} d_h^{\pi^\star}(s,a) b_{h,1}(s,a)}_{(II)},$$

for any $h \in [H]$.

We first study the term (I) of Eq.(4). Observe that for any $(s, a)$,

$$[\widehat{\mathbb{V}}_{h,0} \widehat{V}_{h+1}^{\mathrm{ref}}](s,a) - [\mathbb{V}_h \widehat{V}_{h+1}^{\mathrm{ref}}](s,a)$$
$$= \widehat{\mathbb{P}}_{h,0}(\widehat{V}_{h+1}^{\mathrm{ref}})^2 - (\widehat{\mathbb{P}}_{h,0}\widehat{V}_{h+1}^{\mathrm{ref}})^2 - (\mathbb{P}_h(\widehat{V}_{h+1}^{\mathrm{ref}})^2 - (\mathbb{P}_h\widehat{V}_{h+1}^{\mathrm{ref}})^2) \qquad \text{(``$(s,a)$'' is omitted)}$$
$$\leq |(\widehat{\mathbb{P}}_{h,0} - \mathbb{P}_h)(\widehat{V}_{h+1}^{\mathrm{ref}})^2| + |(\widehat{\mathbb{P}}_{h,0} + \mathbb{P}_h)\widehat{V}_{h+1}^{\mathrm{ref}} \cdot (\widehat{\mathbb{P}}_{h,0} - \mathbb{P}_h)\widehat{V}_{h+1}^{\mathrm{ref}}|$$
$$\leq |(\widehat{\mathbb{P}}_{h,0} - \mathbb{P}_h)(\widehat{V}_{h+1}^{\mathrm{ref}})^2| + 2H|(\widehat{\mathbb{P}}_{h,0} - \mathbb{P}_h)\widehat{V}_{h+1}^{\mathrm{ref}}| \qquad (|\widehat{V}^{\mathrm{ref}}| \leq H)$$
$$\leq c\sqrt{\frac{H^4 \iota}{N_0(s,a) \vee 1}}. \tag{5}$$

where the last inequality follows from the Azuma-Hoeffding inequality and the fact that $\widehat{V}_{h+1}^{\mathrm{ref}}(s)$ is obtained from data independent of $\widehat{P}_{h,0}$.

Thus, we obtain the following bound for term (I)

$$(I) = \sum_{h=1}^H \sum_{(s,a) \in \mathcal{S} \times \mathcal{A}} d_h^{\pi^\star}(s,a) b_{h,0}(s,a)$$

$$= c \sum_{h=1}^H \sum_{(s,a) \in \mathcal{S} \times \mathcal{A}} d_h^{\pi^\star}(s,a) \left( \sqrt{\frac{[\widehat{\mathbb{V}}_{h,0}(\widehat{V}_{h+1}^{\mathrm{ref}})](s,a)\iota}{N_{h,0}(s,a) \vee 1}} + \frac{H\iota}{N_{h,0}(s,a) \vee 1} \right)$$

$$\leq c \sum_{h=1}^H \sum_{(s,a) \in \mathcal{S} \times \mathcal{A}} d_h^{\pi^\star}(s,a) \left( \sqrt{\frac{[\mathbb{V}_h(\widehat{V}_{h+1}^{\mathrm{ref}})](s,a)\iota + \sqrt{\frac{H^4\iota}{N_0(s,a)\vee 1}}}{N_{h,0}(s,a) \vee 1}} + \frac{H\iota}{N_{h,0}(s,a) \vee 1} \right)$$
$$\text{(by Eq.(5))}$$

$$\leq c \sum_{h=1}^H \sum_{(s,a) \in \mathcal{S} \times \mathcal{A}} d_h^{\pi^\star}(s,a) \left( \sqrt{\frac{[\mathbb{V}_h(\widehat{V}_{h+1}^{\mathrm{ref}})](s,a)\iota}{N_{h,0}(s,a) \vee 1}} + \frac{H\iota^{1/4}}{(N_{h,0}(s,a) \vee 1)^{3/4}} + \frac{H\iota}{N_{h,0}(s,a) \vee 1} \right)$$

$$\leq c \sum_{h=1}^H \sum_{(s,a) \in \mathcal{S} \times \mathcal{A}} d_h^{\pi^\star}(s,a) \left( \sqrt{\frac{[\mathbb{V}_h(\widehat{V}_{h+1}^{\mathrm{ref}})](s,a)\iota}{N_{h,0}(s,a) \vee 1}} + \sqrt{\frac{1}{N_{h,0}(s,a) \vee 1}} + \frac{H^2\iota^{1/2} + H\iota}{N_{h,0}(s,a) \vee 1} \right)$$

$$= c \underbrace{\sum_{h=1}^H \sum_{(s,a) \in \mathcal{S} \times \mathcal{A}} d_h^{\pi^\star}(s,a) \sqrt{\frac{[\mathbb{V}_h(\widehat{V}_{h+1}^{\mathrm{ref}})](s,a)\iota}{N_{h,0}(s,a) \vee 1}}}_{(I.a)} + c \underbrace{\sum_{h=1}^H \sum_{(s,a) \in \mathcal{S} \times \mathcal{A}} d_h^{\pi^\star}(s,a) \sqrt{\frac{1}{N_{h,0}(s,a) \vee 1}}}_{(I.b)}$$

$$\tag{6}$$

$$+ c \underbrace{\sum_{h=1}^{H} \sum_{(s,a)\in\mathcal{S}\times\mathcal{A}} d_h^{\pi_\star}(s,a) \frac{H^2 \iota^{1/2} + H\iota}{N_{h,0}(s,a) \vee 1}}_{\text{(I.c)}}$$

where the last inequality follows from Cauchy–Schwarz inequality.

We now discuss the three terms in Eq.(6) separately:

$$\begin{aligned}
\text{(I.a)} &= \sum_{h=1}^{H} \sum_{(s,a)\in\mathcal{S}\times\mathcal{A}} d_h^{\pi_\star}(s,a) \sqrt{\frac{[\mathbb{V}_h(\widehat{V}_{h+1}^{\mathrm{ref}})](s,a)\iota}{N_{h,0}(s,a) \vee 1}} \\
&\leq \sum_{h=1}^{H} \sum_{(s,a)\in\mathcal{S}\times\mathcal{A}} d_h^{\pi_\star}(s,a) \cdot \sqrt{\frac{[\mathbb{V}_h(\widehat{V}_{h+1}^{\mathrm{ref}})](s,a)\iota^2}{n_0 d_h^{\mu}(s,a)}} &&\text{(by the concentration event } \mathcal{E}\text{(iii))} \\
&\leq \sqrt{\frac{C^\star \iota^2}{n_0}} \sum_{h=1}^{H} \sum_{(s,a)\in\mathcal{S}\times\mathcal{A}} \sqrt{d_h^{\pi_\star}(s,a)[\mathbb{V}_h(\widehat{V}_{h+1}^{\mathrm{ref}})](s,a)} \\
&= \sqrt{\frac{C^\star \iota^2}{n_0}} \sum_{h=1}^{H} \sum_{(s,a)\in\mathcal{S}\times\mathcal{A}} \sqrt{\mathbb{1}\{a = \pi_\star(s)\} d_h^{\pi_\star}(s,a)[\mathbb{V}_h(\widehat{V}_{h+1}^{\mathrm{ref}})](s,a)} \\
&\leq \sqrt{\frac{C^\star \iota^2}{n_0}} \sqrt{\sum_{h=1}^{H} \sum_{(s,a)\in\mathcal{S}\times\mathcal{A}} \mathbb{1}\{a = \pi_\star(s)\}} \cdot \sqrt{\sum_{h=1}^{H} \sum_{(s,a)\in\mathcal{S}\times\mathcal{A}} d_h^{\pi_\star}(s,a)[\mathbb{V}_h(\widehat{V}_{h+1}^{\mathrm{ref}})](s,a)} \\
&&&\text{(by Cauchy–Schwarz inequality)} \\
&\leq \sqrt{\frac{HSC^\star \iota^2}{n_0}} \sqrt{\sum_{h=1}^{H} \sum_{(s,a)\in\mathcal{S}\times\mathcal{A}} d_h^{\pi_\star}(s,a)[\mathbb{V}_h(\widehat{V}_{h+1}^{\mathrm{ref}})](s,a)} &&(7) \\
&\leq \sqrt{\frac{HSC^\star \iota^2}{n_0}} \sqrt{H^2 + c\sqrt{\frac{H^9 SC^\star \iota^2}{n_{\mathrm{ref}}}}} &&(8) \\
&&&\text{(follows from the total variance lemma (Lemma C.4))} \\
&\leq \sqrt{\frac{H^3 SC^\star \iota^2}{n_0}} + c\sqrt{\frac{HSC^\star \iota^2}{n_0}} \sqrt[4]{\frac{H^9 SC^\star \iota^2}{n_{\mathrm{ref}}}} \\
&\leq \sqrt{\frac{H^3 SC^\star \iota^2}{n_0}} + c\left(\sqrt{\frac{H^3 SC^\star \iota^2}{n_0}} + \frac{H^4 SC^\star \iota^2}{\sqrt{n_0 n_{\mathrm{ref}}}}\right) &&\text{(by } \sqrt{ab} \leq (a+b)/2) \\
&\leq c\left(\sqrt{\frac{H^3 SC^\star \iota^2}{n_0}} + \frac{H^4 SC^\star \iota^2}{\sqrt{n_0 n_{\mathrm{ref}}}}\right).
\end{aligned}$$

Term (I.b) is a smaller-order term compared with (I.a):

$$\begin{aligned}
\text{(I.b)} &= \sum_{h=1}^{H} \sum_{(s,a)\in\mathcal{S}\times\mathcal{A}} d_h^{\pi_\star}(s,a) \sqrt{\frac{1}{N_{h,0}(s,a) \vee 1}} \\
&\leq \sum_{h=1}^{H} \sum_{(s,a)\in\mathcal{S}\times\mathcal{A}} d_h^{\pi_\star}(s,a) \cdot \sqrt{\frac{\iota}{n_0 d_h^{\mu}(s,a)}} &&\text{(by the concentration event } \mathcal{E}\text{(iii))} \\
&\leq \sqrt{\frac{C^\star \iota}{n_0}} \sum_{h=1}^{H} \sum_{(s,a)\in\mathcal{S}\times\mathcal{A}} \sqrt{d_h^{\pi_\star}(s,a)}
\end{aligned}$$

$$= \sqrt{\frac{C^\star \iota}{n_0}} \sum_{h=1}^{H} \sum_{(s,a)\in\mathcal{S}\times\mathcal{A}} \sqrt{\mathbb{1}\{a = \pi_\star(s)\} d_h^{\pi_\star}(s,a)}$$

$$\leq \sqrt{\frac{C^\star \iota}{n_0}} \sqrt{\sum_{h=1}^{H} \sum_{(s,a)\in\mathcal{S}\times\mathcal{A}} \mathbb{1}\{a = \pi_\star(s)\}} \cdot \sqrt{\sum_{h=1}^{H} \sum_{(s,a)\in\mathcal{S}\times\mathcal{A}} d_h^{\pi_\star}(s,a)}$$

$$\text{(by Cauchy–Schwarz inequality)}$$

$$\leq \sqrt{\frac{H^2 C^\star \iota}{n_0}}. \tag{9}$$

Finally, term (I.c)

$$\text{(I.c)} = \sum_{h=1}^{H} \sum_{(s,a)\in\mathcal{S}\times\mathcal{A}} d_h^{\pi_\star}(s,a) \frac{H^2 \iota^{1/2} + H\iota}{N_{h,0}(s,a) \vee 1}$$

$$\leq \sum_{h=1}^{H} \sum_{(s,a)\in\mathcal{S}\times\mathcal{A}} d_h^{\pi_\star}(s,a) \cdot \frac{H^2 \iota^{3/2} + H\iota^2}{n_0 d_h^\mu(s,a)} \qquad \text{(by the concentration event } \mathcal{E}\text{(iii))}$$

$$\leq \frac{H^3 S C^\star \iota^{3/2} + H^2 S C^\star \iota^2}{n_0}. \tag{10}$$

Substituting Eq.(8), Eq.(9), and Eq.(10) into Eq.(6), we obtain

$$\text{(I)} = \sum_{h=1}^{H} \sum_{(s,a)\in\mathcal{S}\times\mathcal{A}} d_h^{\pi_\star}(s,a) b_{h,0}(s,a)$$

$$\leq c \cdot \left( \sqrt{\frac{H^3 S C^\star \iota^2}{n_0}} + \frac{H^4 S C^\star \iota^2}{\sqrt{n_0 n_{\text{ref}}}} + \frac{H^3 S C^\star \iota^{3/2} + H^2 S C^\star \iota^2}{n_0} \right). \tag{11}$$

We now study the term (II) of Eq.(4). Let $g_{h+1} := \widehat{V}_{h+1} - \widehat{V}_{h+1}^{\text{ref}}$ (which by our data splitting only depends on the datasets $\mathcal{D}_{\text{ref}}, \mathcal{D}_0$ and is independent of $\mathcal{D}_{1,h}$). By a similar argument as Eq.(5), we have

$$[\widehat{\mathbb{V}}_{h,1} g](s,a) - [\mathbb{V}_h g](s,a) \leq c\sqrt{\frac{H^4 \iota}{N_1(s,a) \vee 1}}.$$

for any $(s,a)$. Thus,

$$\text{(II)} = \sum_{h=1}^{H} \sum_{(s,a)\in\mathcal{S}\times\mathcal{A}} d_h^{\pi_\star}(s,a) b_{h,1}(s,a)$$

$$= c \sum_{h=1}^{H} \sum_{(s,a)\in\mathcal{S}\times\mathcal{A}} d_h^{\pi_\star}(s,a) \left( \sqrt{\frac{\left[\widehat{\mathbb{V}}_h(\widehat{V}_{h+1} - \widehat{V}_{h+1}^{\text{ref}})\right](s,a)\iota}{N_{h,1}(s,a) \vee 1}} + \frac{H\iota}{N_{h,0}(s,a) \vee 1} \right)$$

$$\leq c \sum_{h=1}^{H} \sum_{(s,a)\in\mathcal{S}\times\mathcal{A}} d_h^{\pi_\star}(s,a) \left( \sqrt{\frac{\left[\mathbb{V}_h(\widehat{V}_{h+1} - \widehat{V}_{h+1}^{\text{ref}})\right](s,a)\iota + \sqrt{\frac{H^4\iota}{N_1(s,a)\vee 1}}}{N_{h,1}(s,a) \vee 1}} + \frac{H\iota}{N_{h,0}(s,a) \vee 1} \right)$$

$$\leq c \sum_{h=1}^{H} \sum_{(s,a)\in\mathcal{S}\times\mathcal{A}} d_h^{\pi_\star}(s,a) \left( \sqrt{\frac{\left[\mathbb{V}_h(\widehat{V}_{h+1} - \widehat{V}_{h+1}^{\text{ref}})\right](s,a)\iota}{N_{h,1}(s,a) \vee 1}} + \sqrt{\frac{1}{N_{h,1}(s,a) \vee 1}} + \frac{H^2\iota^{1/2} + H\iota}{N_{h,1}(s,a) \vee 1} \right)$$

(by a similar argument of Eq.(6))

$$
= c \sum_{h=1}^{H} \sum_{(s,a)\in\mathcal{S}\times\mathcal{A}} d_h^{\pi_\star}(s,a) \underbrace{\sqrt{\frac{\left[\mathbb{V}_h(\widehat{V}_{h+1} - \widehat{V}_{h+1}^{\mathrm{ref}})\right](s,a)\iota}{N_{h,1}(s,a)\vee 1}}}_{\text{(II.a)}}
\tag{12}
$$

$$
+ c \sum_{h=1}^{H} \sum_{(s,a)\in\mathcal{S}\times\mathcal{A}} d_h^{\pi_\star}(s,a) \underbrace{\sqrt{\frac{1}{N_{h,1}(s,a)\vee 1}}}_{\text{(II.b)}} + c \sum_{h=1}^{H} \sum_{(s,a)\in\mathcal{S}\times\mathcal{A}} d_h^{\pi_\star}(s,a) \underbrace{\frac{H^2\iota^{1/2} + H\iota}{N_{h,1}(s,a)\vee 1}}_{\text{(II.c)}}.
$$

Now, we also bound (II.a), (II.b) and (II.c) separately:

$$
\text{(II.a)} = \sum_{h=1}^{H} \sum_{(s,a)\in\mathcal{S}\times\mathcal{A}} d_h^{\pi_\star}(s,a)\sqrt{\frac{\left[\mathbb{V}_h(\widehat{V}_{h+1} - \widehat{V}_{h+1}^{\mathrm{ref}})\right](s,a)\iota}{N_{h,1}(s,a)\vee 1}}
$$

$$
\le \sum_{h=1}^{H} \sum_{(s,a)\in\mathcal{S}\times\mathcal{A}} d_h^{\pi_\star}(s,a)\cdot\sqrt{\frac{H\left[\mathbb{V}_h(\widehat{V}_{h+1} - \widehat{V}_{h+1}^{\mathrm{ref}})\right](s,a)\iota^2}{n_1 d_h^\mu(s,a)}}
$$

(by the concentration event $\mathcal{E}$(iv))

$$
\le \sqrt{\frac{HC^\star\iota^2}{n_1}} \sum_{h=1}^{H} \sum_{(s,a)\in\mathcal{S}\times\mathcal{A}} \sqrt{d_h^{\pi_\star}(s,a)\left[\mathbb{V}_h(\widehat{V}_{h+1} - \widehat{V}_{h+1}^{\mathrm{ref}})\right](s,a)}
$$

$$
= \sqrt{\frac{HC^\star\iota^2}{n_1}} \sum_{h=1}^{H} \sum_{s} \sqrt{d_h^{\pi_\star}(s)\left[\mathbb{V}_h(\widehat{V}_{h+1} - \widehat{V}_{h+1}^{\mathrm{ref}})\right](s,\pi_{\star,h}(s))}
$$

$$
\le \sqrt{\frac{H^2 SC^\star\iota^2}{n_1}} \sum_{h=1}^{H} \sum_{s} d_h^{\pi_\star}(s)\left[\mathbb{V}_h(\widehat{V}_{h+1} - \widehat{V}_{h+1}^{\mathrm{ref}})\right](s,\pi_{\star,h}(s))
$$

(by Cauchy–Schwarz inequality, e.g., similar to Eq.(7))

$$
\le \sqrt{\frac{H^2 SC^\star\iota^2}{n_1}} \sum_{h=1}^{H} \sum_{s} d_h^{\pi_\star}(s)\left[\mathbb{P}_h(\widehat{V}_{h+1} - \widehat{V}_{h+1}^{\mathrm{ref}})^2\right](s,\pi_{\star,h}(s))
$$

$$
\le \sqrt{\frac{H^4 SC^\star\iota^2}{n_1}} \max_{h\in\{1,2,...,H\}} \sum_{s} d_h^{\pi_\star}(s)\left[\mathbb{P}_h(\widehat{V}_{h+1} - \widehat{V}_{h+1}^{\mathrm{ref}})^2\right](s,\pi_{\star,h}(s))
$$

$$
\le \sqrt{\frac{H^4 SC^\star\iota^2}{n_1}} \max_{h\in\{1,2,...,H\}} \sum_{s} d_h^{\pi_\star}(s)\left[\mathbb{P}_h(V_{h+1}^\star - \widehat{V}_{h+1}^{\mathrm{ref}})^2\right](s,\pi_{\star,h}(s))
$$

(by $\widehat{V}_{h+1} \le V_{h+1}^\star$ from Lemma C.2)

$$
= \sqrt{\frac{H^4 SC^\star\iota^2}{n_1}} \max_{h\in\{1,2,...,H\}} \sum_{s'} d_{h+1}^{\pi_\star}(s')\left(V_{h+1}^\star(s') - \widehat{V}_{h+1}^{\mathrm{ref}}(s')\right)^2
$$

$$
\le \sqrt{\frac{H^4 SC^\star\iota^2}{n_1}} \max_{h\in\{1,2,...,H\}} \sum_{s} d_h^{\pi_\star}(s)\left(V_h^\star(s) - \widehat{V}_h^{\mathrm{ref}}(s)\right)\cdot H
$$

$$
\le c\frac{H^{5.5} SC^\star\iota^2}{\sqrt{n_1 n_{\mathrm{ref}}}},
\tag{13}
$$

where the last inequality follows from the guarantee for $\widehat{V}^{\mathrm{ref}}$ in Lemma C.5:

$$
\max_{h\in\{1,2,...,H\}} \sum_{s} d_h^{\pi_\star}(s)(V_h^\star(s) - \widehat{V}_h^{\mathrm{ref}}(s)) \le c\sqrt{\frac{H^5 SC^\star\iota^2}{n_{\mathrm{ref}}}}.
$$

By similar arguments as Eq.(9) and Eq.(10), we also have the bounds on (II.b) and (II.c) as follows:

$$
\begin{aligned}
\text{(II.b)} &= \sum_{h=1}^{H} \sum_{(s,a)\in\mathcal{S}\times\mathcal{A}} d_h^{\pi_\star}(s,a)\sqrt{\frac{1}{N_{h,1}(s,a)\vee 1}} \\
&\leq \sum_{h=1}^{H} \sum_{(s,a)\in\mathcal{S}\times\mathcal{A}} d_h^{\pi_\star}(s,a)\cdot\sqrt{\frac{H\iota}{n_1 d_h^{\mu}(s,a)}} \qquad\text{(by the concentration event } \mathcal{E}\text{(iv))} \\
&\leq \sqrt{\frac{HC^\star\iota}{n_1}}\sum_{h=1}^{H}\sum_{(s,a)\in\mathcal{S}\times\mathcal{A}}\sqrt{d_h^{\pi_\star}(s,a)} \\
&= \sqrt{\frac{HC^\star\iota}{n_1}}\sum_{h=1}^{H}\sum_{(s,a)\in\mathcal{S}\times\mathcal{A}}\sqrt{\mathbb{1}\{a=\pi_\star(s)\}d_h^{\pi_\star}(s,a)} \\
&\leq \sqrt{\frac{HC^\star\iota}{n_1}}\sqrt{\sum_{h=1}^{H}\sum_{(s,a)\in\mathcal{S}\times\mathcal{A}}\mathbb{1}\{a=\pi_\star(s)\}}\cdot\sqrt{\sum_{h=1}^{H}\sum_{(s,a)\in\mathcal{S}\times\mathcal{A}}d_h^{\star}(s,a)} \\
&\qquad\qquad\qquad\qquad\qquad\qquad\qquad\qquad\qquad\text{(by Cauchy–Schwarz inequality)} \\
&\leq \sqrt{\frac{H^3 SC^\star\iota}{n_1}}. 
\end{aligned} \tag{14}
$$

$$
\begin{aligned}
\text{(II.c)} &= \sum_{h=1}^{H} \sum_{(s,a)\in\mathcal{S}\times\mathcal{A}} d_h^{\pi_\star}(s,a)\frac{H^2\iota^{1/2}+H\iota}{N_{h,1}(s,a)\vee 1} \\
&\leq \sum_{h=1}^{H} \sum_{(s,a)\in\mathcal{S}\times\mathcal{A}} d_h^{\pi_\star}(s,a)\cdot\frac{H^3\iota^{3/2}+H^2\iota^2}{n_1 d_h^{\mu}(s,a)} \qquad\text{(by the concentration event } \mathcal{E}\text{(iv))} \\
&\leq \frac{H^4 SC^\star\iota^{3/2}+H^3 SC^\star\iota^2}{n_1}.
\end{aligned} \tag{15}
$$

Substituting Eq.(13), Eq.(14), and Eq.(15) into Eq.(12), we obtain

$$
\begin{aligned}
\text{(II)} &= \sum_{h=1}^{H} \sum_{(s,a)\in\mathcal{S}\times\mathcal{A}} d_h^{\pi_\star}(s,a)b_{h,1}(s,a) \\
&\leq c\cdot\left(\frac{H^{5.5} SC^\star\iota^2}{\sqrt{n_1 n_{\mathrm{ref}}}}+\sqrt{\frac{H^3 SC^\star\iota}{n_1}}+\frac{H^4 SC^\star\iota^{3/2}+H^3 SC^\star\iota^2}{n_1}\right).
\end{aligned} \tag{16}
$$

By definition, we know $n_{\mathrm{ref}} = n_0 = n_1 = n/3$. Therefore, combining Eq.(16) and Eq.(11), we obtain

$$
\begin{aligned}
V_1^\star(s_1)-\widehat{V}_1(s_1) &\leq 2\cdot\text{(I)}+2\cdot\text{(II)} \\
&\leq c\cdot\left(\sqrt{\frac{H^3 SC^\star\iota^2}{n}}+\frac{H^{5.5}SC^\star\iota^2}{n}\right).
\end{aligned}
$$

The right-hand-side is upper bounded by $\varepsilon$ as long as

$$
n \geq \widetilde{O}\left(H^3 SC^\star\iota^2/\varepsilon^2 + \frac{H^{5.5}SC^\star\iota^2}{\varepsilon}\right)
$$

Finally, by the monotonicity property in Lemma C.2, the above also implies the guarantee on $\widehat{\pi}$:

$$
V_1^\star(s_1)-V_1^{\widehat{\pi}}(s_1)\leq c\cdot\left(\sqrt{\frac{H^3 SC^\star\iota^2}{n}}+\frac{H^{5.5}SC^\star\iota^2}{n}\right)\leq\varepsilon.
$$

This completes the proof.

# D Proof of Theorem 3

To avoid notational clash, in this proof we use upper-case letters $S_h, A_h, R_h$ to denote the actual states and actions seen during the algorithm execution (which are random variables), and reserve the lower-case letters $s_i, s_g, s_b, a$ for indexing the (fixed) states and actions of the MDP.

Define an integer

$$K := \min\{\lfloor C^\star \rfloor, A\}.$$

By our assumption that $C^\star \geq 2$, we have $2 \leq K \leq A$ and $K \in [2/3, 1] \cdot \min\{C^\star, A\}$. Therefore, it suffices to prove the desired performance lower bound for $n \leq c_0 \cdot H^3 SK/\varepsilon^2$.

**Construction of hard instances**  We now construct a family of MDPs with $S + 2$ states, $2H + 1$ steps, and $A$ actions for any $S \geq 1$ and $H \geq 1$. (This rescaling only affects $S, H$ by at most a multiplicative constant and thus does not affect our result.)

Each MDP $M_{a^\star}$ is indexed by a vector $a^\star = (a^\star_{h,i}) \in [A]^{HS}$ and is specified as follows:

- State space: There are $S$ "bandit states" $\{s_i\}_{i \in [S]}$, one "good state" $s_g$, and one "bad state" $s_b$.

- The action space is $\mathcal{A} := [A]$.

- Transitions:

  - At each $h \in \{1, \ldots, H\}$, the bandit state $s_i$ can only transition to $s_i$ itself, $s_g$, or $s_b$. The transition probabilities are

$$\begin{cases} \mathbb{P}_h(s_i|s_i, a) = 1 - \dfrac{1}{H} \quad \text{for all } a \in [A], \\[2mm] \mathbb{P}_h(s_g|s_i, a) = \mathbb{P}_h(s_b|s_i, a) = \dfrac{1}{2H} \quad \text{for all } a \neq a^\star_{h,i}, \\[2mm] \mathbb{P}_h(s_g|s_i, a^\star_{h,i}) = \dfrac{1}{H}\left(\dfrac{1}{2} + \tau\right), \quad \mathbb{P}_h(s_b|s_i, a^\star_{h,i}) = \dfrac{1}{H}\left(\dfrac{1}{2} - \tau\right), \end{cases}$$

   where $\tau \leq 1/3$ is a parameter to be determined.
  - At $h \geq H + 1$, all bandit states transit to one of $s_g$ and $s_b$ with probability $1/2$ each.
  - $s_g$ and $s_b$ are absorbing states: $\mathbb{P}_h(s_g|s_g, a) = \mathbb{P}_h(s_b|s_b, a)$ for all $h \in [2H + 1]$ and all $a \in [A]$.

- Initial state distribution is uniform on all bandit states: $S_1 \sim \text{Unif}\{s_i\}_{i \in [S]}$.

- Reward: The bandit states do not receive any reward. The good state and bad state also do not receive any reward at $h \leq H + 1$. Finally, for $h \geq H + 2$, the good state receives reward 1 and the bad state receives reward 0 regardless of the action taken:

$$r_h(s_g, a) = 1 \quad \text{and} \quad r_h(s_b, a) = 0 \quad \text{for } H + 2 \leq h \leq 2H + 1.$$

We also let $M_0$ denote the "null" MDP which has the same construction as the above except that there is no "special" actions $a^\star_{h,i}$, that is,

$$\mathbb{P}_h(s_g|s_i, a) = \mathbb{P}_h(s_b|s_i, a) = \frac{1}{2H} \quad \text{for all } a \in [A].$$

**Policies $\pi_\star$ and $\mu$**  In MDP $M_{a^\star}$, at any bandit state $s_i$ and time step $h \leq H$, the optimal action to take is $a^\star_{h,i}$ since it induces a slightly higher probability of transiting to the "good state" $s_g$ than all other actions. At all other states or time steps, the action does not affect anything (the transition and reward do not depend on the action), so for example we could take $a = 1$. To summarize, the following deterministic policy $\pi_\star$ is an optimal policy for $M_{a^\star}$:

- $\pi_{\star,h}(s_i) = a^\star_{h,i}$ for all $i \in [S]$ and $h \in [H]$.
- $\pi_{\star,h}(s_i) = 1$ for all $h \geq H + 1$;
- $\pi_{\star,h}(s_g) = \pi_{\star,h}(s_b) = 1$ for all $h \in [2H + 1]$.

We define our reference policy $\mu$ as follows

- $\mu_h(a|s_i) = \frac{1}{K}\mathbb{1}\{1 \leq a \leq K\}$ for all $i \in [S]$, and $h \in [H]$.
- $\mu_h(1|s) = 1$ whenever $h \geq H+1$ or $s \in \{s_g, s_b\}$.

The following lemma shows that $\mu$ satisfies $C^\star$-concentrability with respect to $\pi_\star$ as long as all optimal actions $a^\star_{h,i} \in \{1,\ldots,K\}$. The proof of this lemma is deferred to Section D.1.

**Lemma D.1** ($\mu$ satisfies single-policy concentrability). *For any $a^\star \in \{1,\ldots,K\}^{HS}$, in the MDP $M_{a^\star}$, we have*

$$\sup_{h,s,a} \frac{d^{\pi_\star}_h(s,a)}{d^\mu_h(s,a)} \leq K \leq C^\star,$$

*where $\pi_\star$ (the optimal policy for $M_{a^\star}$) and $\mu$ are defined as above.*

Lemma D.1 shows that the following family of problems is indeed a subset of the class (1):

$$\left\{ (M_{a^\star}, \mu) : a^\star \in [K]^{HS} \right\} \subset \mathcal{M}_{C^\star}.$$

We further let $\nu$ denote the uniform (prior) distribution on $[K]^{HS}$, that is, $\nu(a^\star = a_0) = 1/K^{HS}$ for all $a_0 \in [K]^{HS}$.

**$\mu$ is uninformative** Note that in the above family of problems, the reference policy $\mu$ is the same for all MDPs. Therefore $\mu$ is *uninformative* in the sense that the set of all online finetuning algorithms that utilize $\mu$ in its execution is equivalent to the set of all usual online RL algorithms for this particular class of MDPs (which by definition may utilize the additional knowledge that this class of MDPs has a policy $\mu$ with good concentrability). In the following, we assume $\widehat{\pi}$ is any online RL algorithm for this class of MDPs.

Without loss of generality, we further restrict attention to algorithms that output a deterministic policy (i.e. $\widehat{\pi}_h(s) \in [A]$)[6]. For any deterministic policy $\widehat{\pi}$ and any index $a_\star$, define the *bandit best-arm identification* loss for any $(h,i) \in [H] \times [S]$ as

$$\ell_{h,i}(\widehat{\pi}, a_\star) := \mathbb{P}_{a_\star}\left(\widehat{\pi}_h(s_i) \neq a^\star_{h,i}\right),$$

and the *total bandit loss* as

$$L(\widehat{\pi}, a_\star) := \sum_{(h,i)\in[H]\times[S]} \ell_{h,i}(\widehat{\pi}, a_\star) = \mathbb{E}_{a_\star}\left[\#\left\{(h,i) : \widehat{\pi}_h(s_i) \neq a^\star_{h,i}\right\}\right].$$

The loss $L$ measures the expected number of $(h,i)$ pairs on which the algorithm failed to identify the best arm $a^\star_{h,i}$. A large loss will translate to a high suboptimality bound, as we make precise in the following lemma. The proof can be found in Section D.2.

**Lemma D.2.** *For any $a^\star \in [A]^{HS}$ and any algorithm outputing a deterministic policy $\widehat{\pi}$, we have*

$$\mathbb{E}_{a^\star}\left[V^\star_{1,M_{a^\star}} - V^{\widehat{\pi}}_{1,M_{a^\star}}\right] = \sum_{h=1}^{H}\sum_{i=1}^{S} \frac{1}{S}\left(1 - \frac{1}{H}\right)^{h-1} \tau \cdot \ell_{h,i}(\widehat{\pi}, a^\star) \geq \frac{\tau}{3S}L(\widehat{\pi}, a^\star).$$

With Lemma D.2 at hand, establishing lower bound on the expected suboptimality reduces to establishing a lower bound on the total bandit loss $L(\widehat{\pi}, a^\star)$.

**Lower bounding the total bandit loss** Fix any $(h,i)$, and consider the individual bandit loss $\mathbb{E}_{a^\star \sim \nu}[\ell_{h,i}(\widehat{\pi}, a^\star)]$ averaged over the prior $a^\star \sim \nu$. For any algorithm $\widehat{\pi}$, we decompose the algorithm into (1) the data collection algorithm $\mathcal{A}$ from which we collect the observed data $X \sim (\mathbb{P}_{a^\star}, \mathcal{A})$, and (2) the estimator $\widehat{\pi}_h(s_i) = f(X)$ for some measurable function $f : \mathcal{X} \to [A]$. We have

$$\mathbb{E}_{a^\star \sim \nu}[\ell_{h,i}(\widehat{\pi}, a^\star)] \geq \inf_f \mathbb{E}_{a^\star \sim \nu}\mathbb{E}_{X \sim (\mathbb{P}_{a^\star}, \mathcal{A})}\left[a^\star_{h,i} \neq f(X)\right].$$

Fixing $\mathcal{A}$, the $\inf_f$ is taken at the Bayes estimator of $a^\star_{h,i}$, which is a function of the posterior $a^\star_{h,i}|X$ (see, e.g. [33, Theorem 1.1 of Section 4]).

---

[6]This is because we can replace $\mathbb{P}_{a_\star}(\widehat{\pi}_h(s) \neq a^\star_{h,i})$ in our proof for deterministic policies with $\mathbb{E}_{a_\star}\left[1 - \widehat{\pi}_h(a^\star_{h,i}|s)\right]$ for stochastic policies, and the proof will follow analogously.

By construction of our MDP $M_{\boldsymbol{a}^\star}$, the prior distribution of $a_{h,i}^\star$ is uniform in $[K]$, and the only data that reveal information about $a_{h,i}^\star$ (i.e. data whose likelihood is affected by $a_{h,i}^\star$) is the observed transitions from state $s_i$ at step $h$ (i.e. $\{(A_h, S_{h+1}') : S_h = s_i\}$). Therefore, the posterior distribution $a_{h,i}^\star | X$ depends only on $\{(A_h, S_{h+1}') : S_h = s_i\}$ as well. Further, a set of sufficient statistics for this posterior is the visitation counts $\{N_h(s_i, a, s') : a \in [A], s' \in \mathcal{S}\}$, where $N_h(s_i, a, s')$ denote the visitation count of $(s_i, a, s')$ at step $h$ within the data $X$. Therefore, we can restrict the $\inf_f$ to the inf over functions of the visitation counts only (denoted as $g$), and obtain that

$$\mathbb{E}_{\boldsymbol{a}^\star \sim \nu}[\ell_{h,i}(\widehat{\pi}, \boldsymbol{a}^\star)] \geq \inf_g \mathbb{E}_{\boldsymbol{a}^\star \sim \nu} \mathbb{P}_{X \sim (\mathbb{P}_{\boldsymbol{a}^\star}, \mathcal{A})} \big[ a_{h,i}^\star \neq g(\{N_h(s_i, a, s') : a \in [A], s' \in \mathcal{S}\}) \big]$$

$$\geq \inf_g \mathbb{E}_{\boldsymbol{a}^\star \sim \nu} \mathbb{P}_{X \sim (\mathbb{P}_{\boldsymbol{0}}, \mathcal{A})} \big[ a_{h,i}^\star \neq g(\{N_h(s_i, a, s') : a \in [A], s' \in \mathcal{S}\}) \big]$$

$$\quad - \mathbb{E}_{\boldsymbol{a}^\star \sim \nu} \mathrm{TV}(\{N_h(s_i, a, s')\}|_{\mathbb{P}_{\boldsymbol{0}}, \mathcal{A}}, \{N_h(s_i, a, s')\}|_{\mathbb{P}_{\boldsymbol{a}^\star}, \mathcal{A}})$$

$$\overset{(i)}{\geq} \inf_g \mathbb{E}_{\boldsymbol{a}^\star_{-(h,i)} \sim \nu_{-(h,i)}} \frac{1}{K} \sum_{a_{h,i}^\star = 1}^K \mathbb{P}_{X \sim (\mathbb{P}_{\boldsymbol{0}}, \mathcal{A})} \big[ a_{h,i}^\star \neq g(\{N_h(s_i, a, s') : a \in [A], s' \in \mathcal{S}\}) \big]$$

$$\quad - \mathbb{E}_{\boldsymbol{a}^\star \sim \nu} \sqrt{\frac{1}{2} \mathrm{KL}(\{N_h(s_i, a, s')\}|_{\mathbb{P}_{\boldsymbol{0}}, \mathcal{A}} \| \{N_h(s_i, a, s')\}|_{\mathbb{P}_{\boldsymbol{a}^\star}, \mathcal{A}})}$$

$$\overset{(ii)}{\geq} \frac{K-1}{K} - \mathbb{E}_{\boldsymbol{a}^\star \sim \nu} \sqrt{\frac{1}{2} \sum_{a=1}^A \mathbb{E}_{\mathbb{P}_{\boldsymbol{0}}, \mathcal{A}}[N_h(s_i, a)] \cdot \mathrm{KL}(\mathbb{P}_{0,h}(\cdot | s_i, a) \| \mathbb{P}_{\boldsymbol{a}^\star, h}(\cdot | s_i, a))}$$

$$\overset{(iii)}{\geq} \frac{1}{2} - \mathbb{E}_{\boldsymbol{a}^\star \sim \nu} \sqrt{\frac{1}{2} \mathbb{E}_{\mathbb{P}_{\boldsymbol{0}}, \mathcal{A}}\big[ N_h(s_i, a_{h,i}^\star) \big] \cdot \mathrm{KL}\big( \mathbb{P}_{0,h}(\cdot | s_i, a_{h,i}^\star) \| \mathbb{P}_{\boldsymbol{a}^\star, h}(\cdot | s_i, a_{h,i}^\star) \big)}$$

$$\overset{(iv)}{\geq} \frac{1}{2} - \mathbb{E}_{\boldsymbol{a}^\star \sim \nu} \sqrt{\mathbb{E}_{\mathbb{P}_{\boldsymbol{0}}, \mathcal{A}}\big[ N_h(s_i, a_{h,i}^\star) \big] \cdot 2\tau^2/H}$$

$$\overset{(v)}{\geq} \frac{1}{2} - \sqrt{\frac{1}{K} \sum_{a=1}^K \mathbb{E}_{\mathbb{P}_{\boldsymbol{0}}, \mathcal{A}}[N_h(s_i, a)] \cdot 2\tau^2/H}$$

$$\geq \frac{1}{2} - \sqrt{\mathbb{E}_{\mathbb{P}_{\boldsymbol{0}}, \mathcal{A}}[N_h(s_i)] \cdot 2\tau^2/HK}.$$

Above, (i) used Pinsker's inequality; (ii) used the KL divergence decomposition [31, Lemma 15.1], and the fact that $\frac{1}{K} \sum_{a_{h,i}^\star = 1}^K \mathbb{P}_{X \sim (\mathbb{P}_{\boldsymbol{0}}, \mathcal{A})} \big[ a_{h,i}^\star \neq g(\{N_h(s_i, a, s') : a \in [A], s' \in \mathcal{S}\}) \big]$ is at least $(K-1)/K$ for any $g$ (it equals either $(K-1)/K$ or $1$ depending on whether $g \in [K]$); (iii) used our construction that $\mathbb{P}_{\boldsymbol{0}}$ and $\mathbb{P}_{\boldsymbol{a}^\star}$ at step $h$ and state $s_i$ only differ in the arm $a_{h,i}^\star$; (iv) used the bound that

$$\mathrm{KL}\left( \left( 1 - \frac{1}{H}, \frac{1}{2H}, \frac{1}{2H} \right) \middle\| \left( 1 - \frac{1}{H}, \frac{1}{H}\left( \frac{1}{2} + \tau \right), \frac{1}{H}\left( \frac{1}{2} - \tau \right) \right) \right) = \frac{1}{2H} \log \frac{1}{1 - 4\tau^2} \leq 4\tau^2/H$$

for $\tau \in [0, 0.4]$; and finally (v) used Jensen's inequality and the fact that $a_{h,i}^\star \sim \mathrm{Unif}([K])$ under $\nu$.

Summing the preceding bound over all $(h, i)$, we get that for any algorithm $\widehat{\pi}$,

$$\mathbb{E}_{\boldsymbol{a}^\star \sim \nu}[L(\widehat{\pi}, \boldsymbol{a}^\star)] = \sum_{h=1}^H \sum_{i=1}^S \mathbb{E}_{\boldsymbol{a}^\star \sim \nu}[\ell_{h,i}(\widehat{\pi}, \boldsymbol{a}^\star)]$$

$$\geq \frac{HS}{2} - \sum_{h=1}^H \sum_{i=1}^S \sqrt{\mathbb{E}_{\mathbb{P}_{\boldsymbol{0}}, \mathcal{A}}[N_h(s_i)] \cdot 2\tau^2/HK}$$

$$\geq HS\left\{ \frac{1}{2} - \sqrt{\frac{1}{HS} \underbrace{\sum_{h=1}^H \sum_{i=1}^S \mathbb{E}_{\mathbb{P}_{\boldsymbol{0}}, \mathcal{A}}[N_h(s_i)]}_{nH} \cdot 2\tau^2/HK} \right\} = HS\left( \frac{1}{2} - \sqrt{2\tau^2 n/HSK} \right).$$

Therefore, as long as $\sqrt{2\tau^2 n/HSK} \leq 1/4$, i.e. $n \leq HSK/(32\tau^2)$, we have

$$\mathbb{E}_{\boldsymbol{a}^\star \sim \nu}[L(\widehat{\pi}, \boldsymbol{a}^\star)] \geq HS/4.$$

**Bandit loss to MDP suboptimality loss**  By the above lower bound on the average risk, for any algorithm $\widehat{\pi}$, there must exist some instance $\boldsymbol{a}^\star \in [K]^{HS}$ for which $L(\widehat{\pi}, \boldsymbol{a}^\star) \geq HS/4$. On this $\boldsymbol{a}^\star$, by Lemma D.2, we get

$$\mathbb{E}_{\boldsymbol{a}^\star}\left[V^\star_{1,M_{\boldsymbol{a}^\star}} - V^{\widehat{\pi}}_{1,M_{\boldsymbol{a}^\star}}\right] \geq \frac{\tau}{3S} \cdot L(\widehat{\pi}, \boldsymbol{a}^\star) \geq \tau H/12.$$

Finally, for any $\varepsilon \leq 1/12$, take $\tau = 12\varepsilon/H \leq 1/3$, we have the following: as long as

$$n \leq \frac{HSK}{32\tau^2} = \frac{H^3 SK}{32 \cdot 12^2 \varepsilon^2},$$

(which is satisfied if $n \leq c_0 \cdot H^3 S \min\{C^\star, A\}/\varepsilon^2$ for some absolute $c_0 > 0$), we have $\mathbb{E}_{\boldsymbol{a}^\star}\left[V^\star_{1,M_{\boldsymbol{a}^\star}} - V^{\widehat{\pi}}_{1,M_{\boldsymbol{a}^\star}}\right] \geq \varepsilon$. This is the desired result. $\qquad\square$

## D.1 Proof of Lemma D.1

Fix any $\boldsymbol{a}^\star$, we show that $d^{\pi^\star}_h(s,a)/d^\mu_h(s,a) \leq K$ for all $(s,a)$. Note that it suffices to consider actions taken by $\pi_\star$ only.

**Bandit states**  For any bandit state $s \in \{s_i\}_{i\in[S]}$ and any $h \leq H$, by construction of our MDP, we have $d^{\pi^\star}_h(s_i) = d^\mu_h(s_i) = \left(1 - \frac{1}{H}\right)^{h-1}$, $\pi_{\star,h}(a^\star_{h,i}|s_i) = 1$, and $\mu_h(a^\star_{h,i}|s_i) = 1/K$, therefore

$$\frac{d^{\pi^\star}_h(s_i, a^\star_{h,i})}{d^\mu_h(s_i, a^\star_{h,i})} = \frac{d^{\pi^\star}_h(s_i) \cdot \pi_{\star,h}(a^\star_{h,i}|s_i)}{d^\mu_h(s_i) \cdot \mu_h(a^\star_{h,i}|s_i)} = K.$$

At $h = H+1$, we have $d^{\pi^\star}_h(s_i) = d^\mu_h(s_i) = (1 - 1/H)^H$, and both $\pi_\star$ and $\mu$ takes action 1 deterministically, and thus $d^{\pi^\star}_h(s_i, 1)/d^\mu_h(s_i, 1) = 1$. At $h \geq H+2$, we have $d^{\pi^\star}_h(s_i) = d^\mu_h(s_i) = 0$. This verifies the $K$-concentrability for all $s_i$.

**Good state and bad state**  For the good state $s_g$ and bad state $s_b$, since both $\pi_\star$ and $\mu$ takes deterministic action 1 at all $h$, it suffices to bound distribution ratio $d^{\pi^\star}_h(s_{\{g,b\}})/d^\mu_h(s_{\{g,b\}})$ over the states only (instead of joint state-actions).

Recall that $\pi_\star$ always takes the optimal action $a^\star_{h,i}$ which leads to $1/H \cdot (1/2 + \tau)$ transition probability to the "good state" $s_g$. Thus at any $h \leq H+1$, we have

$$d^{\pi^\star}_h(s_g) = \sum_{h'=1}^{h-1} \sum_{i=1}^{S} \frac{1}{S}\left(1 - \frac{1}{H}\right)^{h'-1} \cdot \frac{1}{H}\left(\frac{1}{2} + \tau\right) = \left(\sum_{h'=1}^{h-1}\left(1 - \frac{1}{H}\right)^{h'-1} \cdot \frac{1}{H}\right) \cdot \left(\frac{1}{2} + \tau\right).$$

In contrast, $\mu$ only takes the optimal action $a^\star_{h,i}$ with probability $1/K$, and thus we have

$$d^\mu_h(s_g) = \left(\sum_{h'=1}^{h-1}\left(1 - \frac{1}{H}\right)^{h'-1} \cdot \frac{1}{H}\right) \cdot \left(\frac{1}{2} + \frac{\tau}{K}\right).$$

Therefore

$$\frac{d^{\pi^\star}_h(s_g)}{d^\mu_h(s_g)} = \frac{1/2 + \tau}{1/2 + \tau/K} \leq \frac{1/2 + \tau}{1/2} \leq 2 \leq K.$$

Similarly, for the "bad state" $s_b$ we have

$$\frac{d^{\pi^\star}_h(s_b)}{d^\mu_h(s_b)} = \frac{1/2 - \tau}{1/2 - \tau/K} \leq 1 \leq K.$$

For $h \geq H+2$, we have

$$\frac{d^{\pi^\star}_h(s_g)}{d^\mu_h(s_g)} = \frac{d^{\pi^\star}_{H+1}(s_g) + \frac{1}{2} \cdot (1 - 1/H)^H}{d^\mu_{H+1}(s_g) + \frac{1}{2} \cdot (1 - 1/H)^H} \leq 2 \leq K,$$

and similarly

$$\frac{d^{\pi^\star}_h(s_b)}{d^\mu_h(s_b)} = \frac{d^{\pi^\star}_{H+1}(s_b) + \frac{1}{2} \cdot (1 - 1/H)^H}{d^\mu_{H+1}(s_b) + \frac{1}{2} \cdot (1 - 1/H)^H} \leq 1 \leq K,$$

This verifies the $K$-concentrability for $(s_g, s_b)$ as well. $\qquad\square$

## D.2 Proof of Lemma D.2

Fix any $\boldsymbol{a}^\star \in [A]^{HS}$, by construction of our MDP $M_{\boldsymbol{a}^\star}$, only the good state $s_g$ receives a $+1$ reward starting at $h \in \{H+2, \dots, 2H+1\}$. Along each trajectory, there will be exactly one transition from the bandit states $\{s_i\}$ to one of $(s_g, s_b)$. This transition can happen at step $h \leq H$ and state $s_i$ with probability $1/H \cdot (1/2 + \tau)$ if the optimal action $a^\star_{h,i}$ is taken, or probability $1/(2H)$ if any other action is taken. The transition can also happen at step $h = H+1$ but with the same transition probability regardless of the policy. Further, note that the state distribution $d^\pi_h(s_i) = 1/S \cdot (1 - 1/H)^{h-1} =: d_h(s_i)$ (for $h \leq H$) does not depend on the policy $\pi$. Therefore, we have

$$
V^\star_{1, M_{\boldsymbol{a}^\star}} - V^{\widehat{\pi}}_{1, M_{\boldsymbol{a}^\star}}
$$
$$
= \sum_{h=1}^{H} \sum_{i=1}^{S} d_h(s_i) \cdot \left[ \frac{1}{H}\left(\frac{1}{2} + \tau\right) - \frac{1}{2H} \right] \cdot \mathbb{1}\left\{\widehat{\pi}_h(s_i) \neq a^\star_{h,i}\right\} \cdot H
$$
$$
= \sum_{h=1}^{H} \sum_{i=1}^{S} \frac{1}{S}\left(1 - \frac{1}{H}\right)^{h-1} \tau \cdot \mathbb{1}\left\{\widehat{\pi}_h(s_i) \neq a^\star_{h,i}\right\}.
$$

Taking expectation with respect to the algorithm execution within the MDP $M_{\boldsymbol{a}^\star}$, we get

$$
\mathbb{E}_{\boldsymbol{a}^\star}\left[ V^\star_{1, M_{\boldsymbol{a}^\star}} - V^{\widehat{\pi}}_{1, M_{\boldsymbol{a}^\star}} \right] = \sum_{h=1}^{H} \sum_{i=1}^{S} \frac{1}{S}\left(1 - \frac{1}{H}\right)^{h-1} \tau \cdot \underbrace{\mathbb{P}_{\boldsymbol{a}^\star}\left(\widehat{\pi}_h(s_i) \neq a^\star_{h,i}\right)}_{\ell_{h,i}(\widehat{\pi}, \boldsymbol{a}^\star)}
$$
$$
= \sum_{h=1}^{H} \sum_{i=1}^{S} \frac{1}{S} \underbrace{\left(1 - \frac{1}{H}\right)^{h-1}}_{\geq (1-1/H)^{H-1} \geq e^{-1} \geq 1/3} \tau \cdot \ell_{h,i}(\widehat{\pi}, \boldsymbol{a}^\star)
$$
$$
\geq \frac{\tau}{3S} \cdot \sum_{h=1}^{H} \sum_{i=1}^{S} \ell_{h,i}(\widehat{\pi}, \boldsymbol{a}^\star) = \frac{\tau}{3S} \cdot L(\widehat{\pi}, \boldsymbol{a}^\star).
$$

This proves the lemma. $\qquad\square$

# E Proofs for Section 5

## E.1 Algorithm UCBVI-UpLow

We present the UCBVI-UpLow algorithm in Algorithm 4. This algorithm is a specialization of the Nash-VI algorithm of [34] (originally developed for two-player Markov games) into the case with a single player, and with additional modifications on the output value functions and policy that are similar to the certified policy technique of [7]. Above, the Bonus function on Line 7 is taken as the Bernstein bonus:

$$
\text{Bonus}(t, \widehat{\sigma}^2) := c\left( \sqrt{\widehat{\sigma}^2 \iota / t} + (H - h_\star)^2 S\iota / t \right),
$$

where $\iota := \log(HSAn/\delta)$ is a log factor and $c > 0$ is some absolute constant.

We now present the main guarantee for UCBVI-UpLow, which is going to be used in proving the main theorem.

**Lemma E.1** (Theoretical guarantee for UCBVI-UpLow). *Suppose Algorithm 4 is run for $n_{\text{UCB}}$ episodes. Then, the output lower value estimate $\underline{V}^{\text{out}}_{h_\star+1}$ and the policy $\pi^{\text{out}}_{h_\star+1:H}$ satisfies the following with probability at least $1 - \delta$:*

*(a) Small error in lower value estimates: $0 \leq \underline{V}^{\text{out}}_{h_\star+1}(s) \leq V^\star_{h_\star+1}(s)$ for all $s \in \mathcal{S}$, and*

$$
\sum_{s \in \mathcal{S}} d_{h_\star+1}(s)\left(V^\star_{h_\star+1}(s) - \underline{V}^{\text{out}}_{h_\star+1}(s)\right) \leq O\left( \sqrt{\frac{(H - h_\star)^3 SA\iota^3}{n_{\text{UCB}}}} + \frac{(H - h_\star)^3 S^2 A\iota^3}{n_{\text{UCB}}} \right),
$$

*where $d_{h_\star+1}(\cdot)$ is the distribution of the initial state $s_{h_\star+1}$.*

**Algorithm 4** UCB-VI with Upper and Lower Confidence Bounds (UCBVI-UPLOW)

**Require:** Starting time step $h_\star + 1$, end time step $H$, number of episodes $n_{\mathrm{UCB}}$.

1: **Initialize:** For any $(s, a, h, s')$: $\overline{Q}_h(s, a) \leftarrow H - h_\star$, $\underline{Q}_h(s, a) \leftarrow 0$, $N_h(s) = N_h(s, a) = N_h(s, a, s') \leftarrow 0$.

2: **for** Episode $k = 1, \ldots, n_{\mathrm{UCB}}$ **do**

3:     **for** step $h = H, \ldots, h_\star + 1$ **do**

4:         **for** $(s, a) \in \mathcal{S} \times \mathcal{A}$ **do**

5:             $t \leftarrow N_h(s, a)$.

6:             **if** $t > 0$ **then**

7:                 $\beta \leftarrow \mathrm{BONUS}(t, \widehat{\mathbb{V}}_h[(\overline{V}_{h+1} + \underline{V}_{h+1})/2](s, a))$

8:                 $\gamma \leftarrow c/(H - h_\star) \cdot \widehat{\mathbb{P}}_h(\overline{V}_{h+1} - \underline{V}_{h+1})(s, a)$.

9:                 $\overline{Q}_h(s, a) \leftarrow \min \left\{ (r_h + \widehat{\mathbb{P}}_h \overline{V}_{h+1})(s, a) + \gamma + \beta, H - h_\star \right\}$.

10:                $\underline{Q}_h(s, a) \leftarrow \max \left\{ (r_h + \widehat{\mathbb{P}}_h \underline{V}_{h+1})(s, a) - \gamma - \beta, 0 \right\}$

11:             **end if**

12:         **end for**

13:         **for** $s \in \mathcal{S}$ **do**

14:             $\pi_h(s) \leftarrow \arg\max_{a \in \mathcal{A}} \overline{Q}_h(s, a)$.

15:             $\overline{V}_h(s) \leftarrow \overline{Q}_h(s, \pi_h(s)); \underline{V}_h(s) \leftarrow \underline{Q}_h(s, \pi_h(s))$.

16:         **end for**

17:     **end for**

18:     Receive an initial state $s_{h_\star + 1}$ from the MDP.

19:     **for** step $h = h_\star + 1, \ldots, H$ **do**

20:         Take action $a_h = \pi_h(s_h)$, observe reward $r_h$ and next state $s_{h+1}$.

21:         Add 1 to $N_h(s_h)$, $N_h(s_h, a_h)$, and $N_h(s_h, a_h, s_{h+1})$.

22:         $\widehat{\mathbb{P}}_h(\cdot | s_h, a_h) \leftarrow N_h(s_h, a_h, \cdot)/N_h(s_h, a_h)$.

23:     **end for**

24: **end for**

25: Let $(\overline{V}_h^k, \underline{V}_h^k, \pi^k)$ denote the value estimates and policy at the beginning of episode $k$.

26: **for** $s \in \mathcal{S}$ **do**

27:     $\overline{V}_{h_\star+1}^{\mathrm{out}}(s) \leftarrow \frac{1}{N_{h_\star+1}(s)} \sum_{k : s_{h_\star+1}^k = s} \overline{V}_{h_\star+1}^k(s)$.

28:     $\underline{V}_{h_\star+1}^{\mathrm{out}}(s) \leftarrow \frac{1}{N_{h_\star+1}(s)} \sum_{k : s_{h_\star+1}^k = s} \underline{V}_{h_\star+1}^k(s)$.

29:     Let policy $\pi_{h_\star+1:H}^{(s)}$ be the uniform mixture of $\pi_{h_\star+1:H}^k$ over all $\left\{ k : s_{h_\star+1}^k = s \right\}$.

30: **end for**

31: **return** Value estimates $(\overline{V}_{h_\star+1}^{\mathrm{out}}, \underline{V}_{h_\star+1}^{\mathrm{out}})$.

        Mixture policy $\pi_{h_\star+1:H}^{\mathrm{out}}$ defined as follows: Play policy $\pi_{h_\star+1:H}^{(s)}$ if $s_{h_\star+1} = s$.

---

*(b)* $\pi^{\mathrm{out}}$ *achieves at least* $\underline{V}^{\mathrm{out}}$ *value: we have* $V_{h_\star+1}^{\pi^{\mathrm{out}}}(s) \geq \underline{V}_{h_\star+1}^{\mathrm{out}}(s)$ *for all* $s \in \mathcal{S}$.

*Proof.* First notice that Algorithm 4 is a special case of the NASH-VI algorithm for two-player Markov games [34, Algorithm 1], where the number of actions for the min-player is one so that the Markov game reduces to a single-player MDP, and the game starts at step $h_\star + 1$ (so that the horizon length is $H - h_\star$ instead of $H$).

Therefore, by [34, Theorem 4], with probability at least $1 - \delta/2$ the following happens

$$\sum_{k=1}^{n_{\mathrm{UCB}}} \overline{V}_{h_\star+1}^k(s_{h_\star+1}^k) - \underline{V}_{h_\star+1}^k(s_{h_\star+1}^k) \leq \sqrt{(H - h_\star)^3 SA n_{\mathrm{UCB}} \iota} + (H - h_\star)^3 S^2 A \iota^2,$$

where $\iota := \log(HSA/\delta)$. Further, [34, Lemma 22] implies that for any $(s, k)$,

$$\overline{V}_{h_\star+1}^k(s) \geq V_{h_\star+1}^\star(s) \geq V_{h_\star+1}^{\pi^k}(s) \geq \underline{V}_{h_\star+1}^k(s)$$

on the same good event. Using the relation betwen $\overline{V}^k_{h_\star+1}$ and $V^\star_{h_\star+1}$ yields

$$\sum_{s\in\mathcal{S}} N_{h_\star+1}(s)\big(V^\star_{h_\star+1}(s) - \underline{V}^{\mathrm{out}}_{h_\star+1}(s)\big)$$

$$\overset{(i)}{=} \sum_{k=1}^{n_{\mathrm{UCB}}} V^\star_{h_\star+1}(s^k_{h_\star+1}) - \underline{V}^k_{h_\star+1}(s^k_{h_\star+1}) \le \sqrt{(H-h_\star)^3 SA n_{\mathrm{UCB}}\iota} + (H-h_\star)^3 S^2 A\iota^2,$$

where (i) used the definition of our $\underline{V}^{\mathrm{out}}$ in Line 28. Now, since $N_{h_\star+1}(s) \sim \mathrm{Bin}(n_{\mathrm{UCB}}, d_{h_\star+1}(s))$, by Lemma A.1 and a union bound over $s\in\mathcal{S}$, we have with probability at least $1-\delta/2$ that

$$N_{h_\star+1}(s) \vee 1 \ge \frac{1}{C\iota}\cdot n_{\mathrm{UCB}} d_{h_\star+1}(s)$$

simultaneously for all $s\in\mathcal{S}$.

Plugging this into the preceding bound further yields

$$\sum_{s\in\mathcal{S}} d_{h_\star+1}(s)\big(V^\star_{h_\star+1}(s) - \underline{V}^{\mathrm{out}}_{h_\star+1}(s)\big)$$

$$\le \frac{C\iota}{n_{\mathrm{UCB}}} \sum_{s\in\mathcal{S}} \big(N_{h_\star+1}(s)\vee 1\big)\cdot\big(V^\star_{h_\star+1}(s) - \underline{V}^{\mathrm{out}}_{h_\star+1}(s)\big)$$

$$\le \frac{C\iota}{n_{\mathrm{UCB}}} \sum_{s\in\mathcal{S}} N_{h_\star+1}(s)\cdot\big(V^\star_{h_\star+1}(s) - \underline{V}^{\mathrm{out}}_{h_\star+1}(s)\big) + \frac{C\iota}{n_{\mathrm{UCB}}} \sum_{s\in\mathcal{S}} \mathbb{1}\{N_{h_\star+1}(s)=0\}\cdot\underbrace{\big(V^\star_{h_\star+1}(s) - \underline{V}^{\mathrm{out}}_{h_\star+1}(s)\big)}_{\le H-h_\star}$$

$$\le C\sqrt{\frac{(H-h_\star)^3 SA\iota^3}{n_{\mathrm{UCB}}}} + C\cdot\frac{(H-h_\star)^3 S^2 A\iota^3 + (H-h_\star)S\iota}{n_{\mathrm{UCB}}}$$

$$\le C\left(\sqrt{\frac{(H-h_\star)^3 SA\iota^3}{n_{\mathrm{UCB}}}} + \frac{(H-h_\star)^3 S^2 A\iota^3}{n_{\mathrm{UCB}}}\right)$$

This shows part (a).

For part (b), by our definition of $\pi^{\mathrm{out}}$ (Line 31), we have for any $s\in\mathcal{S}$ that

$$V^{\pi^{\mathrm{out}}}_{h_\star+1}(s) = V^{\pi^{(s)}}_{h_\star+1}(s)$$

$$= \frac{1}{N_{h_\star+1}(s)} \sum_{k:s^k_{h_\star+1}=s} V^{\pi^k}_{h_\star+1}(s)$$

$$\ge \frac{1}{N_{h_\star+1}(s)} \sum_{k:s^k_{h_\star+1}=s} \underline{V}^k_{h_\star+1}(s) = \underline{V}^{\mathrm{out}}_{h_\star+1}(s).$$

This shows part (b). $\qquad\square$

## E.2  Algorithm TRUNCATED-PEVI-ADV

The algorithm TRUNCATED-PEVI-ADV is similar as the PEVI-ADV algorithm (Algorithm 1) except that the algorithm uses a plug-in estimate of $V^\star_{h_\star+1}$, and only performs (pessimistic) value iteration within step 1 to $h_\star$. For completeness, we describe the algorithm in Algorithm 5.

## E.3  Proof of Theorem 4

We are now ready to present the proof of Theorem 4. Throughout this proof, $C$ denotes an absolute constant that can vary from line to line, and all "good events" happen with probability at least $1-\delta/10$, which combine to yield the $1-\delta$ high probability guarantee for the final bound.

---

**Algorithm 5** Truncated-PEVI-ADV

---

**Require:** Offline dataset $\mathcal{D} = \left\{ (s_1^{(i)}, a_1^{(i)}, r_1^{(i)}, \ldots, s_H^{(i)}, a_H^{(i)}, r_H^{(i)}) \right\}_{i=1}^{n_{\text{offline}}}$. End time step $h_\star$. Value

function $V_{h_\star+1}^{\text{init}}$.

1: Split the dataset $\mathcal{D}$ into $\mathcal{D}_{\text{ref}}$, $\mathcal{D}_0$ and $\{\mathcal{D}_{h,1}\}_{h=1}^{h_\star}$ uniformly at random:

$$n_{\text{ref}} := |\mathcal{D}_{\text{ref}}| = n_{\text{offline}}/3, \;\; n_0 := |\mathcal{D}_0| = n_{\text{offline}}/3, \;\; n_{1,h} := |\mathcal{D}_{h,1}| := n_{\text{offline}}/(3h_\star).$$

2: Learn a reference value function $\widehat{V}_{1:h_\star}^{\text{ref}} \leftarrow$ VI-LCB$(\mathcal{D}_{\text{ref}})$ via of **a truncated version of** VI-LCB(Algorithm 3), with the modification that only the datae from steps $1 : h_\star$ are used, and $\widehat{V}_{h_\star+1}^{\text{ref}} \leftarrow V_{h_\star+1}^{\text{init}}$ **and is not updated**.

3: Let $N_{h,0}(s,a)$ and $N_{h,0}(s,a,s')$ denote the visitation count of $(s,a)$ and $(s,a,s')$ at step $h$ within dataset $\mathcal{D}_0$. Construct empirical model estimates:

$$\widehat{r}_{h,0}(s,a) \leftarrow r_h(s,a)\mathbb{1}\left\{N_{h,0}(s,a) \geq 1\right\},$$

$$\widehat{\mathbb{P}}_{h,0}(s'|s,a) \leftarrow \frac{N_{h,0}(s,a,s')}{N_{h,0}(s,a) \vee 1}.$$

Similarly define $N_{h,1}(s,a)$, $N_{h,1}(s,a,s')$, $(\widehat{r}_{h,1}, \widehat{\mathbb{P}}_{h,1})$ for all $h \in [h_\star]$ based on dataset $\mathcal{D}_{h,1}$.

4: For all $(h,s,a)$, set $b_{h,0}(s,a) \leftarrow c \cdot \left( \sqrt{\frac{[\widehat{\mathbb{V}}_{h,0}\widehat{V}_{h+1}^{\text{ref}}](s,a)\iota}{N_{h,0}(s,a)\vee 1}} + \frac{H\iota}{N_{h,0}(s,a)\vee 1} \right)$, where $\iota :=$ $\log(HSA/\delta)$.

5: Set $\widehat{V}_{h_\star+1}(s) \leftarrow V_{h_\star+1}^{\text{init}}(s)$ for all $s \in \mathcal{S}$. (Note that $\widehat{V}_{h_\star+1}(s)$ **is not updated in the following**.)

6: **for** $h = h_\star, \ldots, 1$ **do**

7:     Set $b_{h,1}(s,a) \leftarrow c \cdot \left( \sqrt{\frac{[\widehat{\mathbb{V}}_{h,1}(\widehat{V}_{h+1} - \widehat{V}_{h+1}^{\text{ref}})](s,a)\iota}{N_{h,1}(s,a)\vee 1}} + \frac{H\iota}{N_{h,1}(s,a)\vee 1} \right)$.

8:     Perform value update for all $(s,a)$:

$$\widehat{Q}_h(s,a) \leftarrow \widehat{r}_{h,0}(s,a) + \left[\widehat{\mathbb{P}}_{h,0}\widehat{V}_{h+1}^{\text{ref}}\right](s,a) - b_{h,0}(s,a) + \left[\widehat{\mathbb{P}}_{h,1}(\widehat{V}_{h+1} - \widehat{V}_{h+1}^{\text{ref}})\right](s,a) - b_{h,1}(s,a);$$

$$\widehat{V}_h(s) \leftarrow \left[\max_a \widehat{Q}_h(s,a)\right] \vee 0.$$

9:     Set $\widehat{\pi}_h(s) \leftarrow \arg\max_a \widehat{Q}_h(s,a)$ for all $s \in \mathcal{S}$.

10: **end for**

11: **return** Policy $\widehat{\pi} = \{\widehat{\pi}_h\}_{1 \leq h \leq h_\star}$.

---

**Guarantees for learned values** First, Stage 1 in our Algorithm 2 runs the UCBVI-UPLOW algorithm with $n_{\text{UCB}} = n/2$ episodes with initial state $s_{h_\star+1} \sim d_{h_\star+1}^\mu$. Therefore by Lemma E.1, its output lower value estimate $\underline{V}_{h_\star+1}$ satisfies $\underline{V}_{h_\star+1}(s) \leq V_{h_\star+1}^\star(s)$ for all $s \in \mathcal{S}$, and

$$\sum_{s \in \mathcal{S}} d_{h_\star+1}^\mu(s)\left(V_{h_\star+1}^\star(s) - \underline{V}_{h_\star+1}(s)\right) \leq C\left( \sqrt{\frac{(H-h_\star)^3 SA\iota^3}{n}} + \frac{(H-h_\star)^3 S^2 A\iota^3}{n} \right).$$

Since $\mu$ satisfies $C^{\text{partial}}$ partial concentratbility for steps $1 : h_\star$ (by Assumption B), it also satisfies the *state-wise* concentrability at step $h_\star + 1$: For any $s' \in \mathcal{S}$ we have

$$\frac{d_{h_\star+1}^{\pi_\star}(s')}{d_{h_\star+1}^\mu(s')} = \frac{\sum_{s,a} d_{h_\star}^{\pi_\star}(s,a)\mathbb{P}_{h_\star}(s'|s,a)}{\sum_{s,a} d_{h_\star}^\mu(s,a)\mathbb{P}_{h_\star}(s'|s,a)} \leq C^{\text{partial}}.$$

Applying this in the preceding bound, we get

$$\sum_{s \in \mathcal{S}} d_{h_\star+1}^{\pi_\star}(s)\left(V_{h_\star+1}^\star(s) - \underline{V}_{h_\star+1}(s)\right) \leq C \cdot C^{\text{partial}}\left( \sqrt{\frac{(H-h_\star)^3 SA\iota^3}{n}} + \frac{(H-h_\star)^3 S^2 A\iota^3}{n} \right) =: \varepsilon_0.$$

Next, in stage 2 we run the TRUNCATED-PEVI-ADV algorithm. In its first (sub)-stage, we learn the reference value function $\widehat{V}^{\mathrm{ref}}$ via a truncated version of the VI-LCB algorithm (which runs for $(n - n_{\mathrm{UCB}})/3 = n/6$ episodes). Since we set $\widehat{V}^{\mathrm{ref}}_{h_\star+1} \leftarrow \underline{V}_{h_\star+1}$ and do not update it, we can imitate the proof of Theorem 1 for steps 1 to $h_\star$ (which uses the $C^{\mathrm{partial}}$ partial-concentrability assumed in Assumption B) and obtain $\widehat{V}^{\mathrm{ref}}_h(s) \leq V^\star_h(s)$ for all $h \in [h_\star]$, $s \in \mathcal{S}$, and (by replacing $H$ with $h_\star$ at appropriate places in the proof in Section B.3)

$$\max_{1 \leq h \leq h_\star} \sum_{s \in \mathcal{S}} d^{\pi_\star}_h(s)\Big(V^\star_h(s) - \widehat{V}^{\mathrm{ref}}_h(s)\Big)$$

$$\leq \sum_{h=1}^{h_\star} \sum_{s,a} d^{\pi_\star}_h(s,a) \cdot b_h(s,a) + \sum_{s \in \mathcal{S}} d^{\pi_\star}_{h_\star+1}(s)\big(V^\star_{h_\star+1}(s) - \underline{V}_{h_\star+1}(s)\big)$$

$$\leq C\sqrt{\frac{H^2 h_\star^3 S C^{\mathrm{partial}} \iota^2}{n}} + \sum_{s \in \mathcal{S}} d^{\pi_\star}_{h_\star+1}(s)\big(V^\star_{h_\star+1}(s) - \underline{V}_{h_\star+1}(s)\big)$$

$$\leq C\sqrt{\frac{H^2 h_\star^3 S C^{\mathrm{partial}} \iota^2}{n}} + \varepsilon_0 =: \varepsilon_1.$$

In its second sub-stage, we learn the final value function $\widehat{V}$ in a similar fashion as the reference-advantage updates in Algorithm 1 (which runs for $n_0 + n_1$ episodes where $n_0 = n_1 = n/6$). Notice again we set $\widehat{V}_{h_\star+1} \leftarrow \underline{V}_{h_\star+1}$ and do not update it. Therefore we can imiate the proof of Theorem 2 (again, this uses the $C^{\mathrm{partial}}$ partial-concentrability assumed in Assumption B) to obtain $\widehat{V}_h(s) \leq V^\star_h(s)$ for all $h \in [h_\star]$, $s \in \mathcal{S}$ and (by replacing $H$ with $h_\star$ at appropriate places in Section C.2)

$$\max_{1 \leq h \leq h_\star} \sum_{s \in \mathcal{S}} d^{\pi_\star}_h(s)\Big(V^\star_h(s) - \widehat{V}_h(s)\Big)$$

$$\leq \sum_{h=1}^{h_\star} \sum_{s,a} d^{\pi_\star}_h(s,a) \cdot (b_{h,0}(s,a) + b_{h,1}(s,a)) + \sum_{s \in \mathcal{S}} d^{\pi_\star}_{h_\star+1}(s)\big(V^\star_{h_\star+1}(s) - \underline{V}_{h_\star+1}(s)\big)$$

$$\leq C\Bigg(\sqrt{\frac{H^2 h_\star S C^{\mathrm{partial}} \iota^2}{n}} + \frac{H^2 h_\star S C^{\mathrm{partial}} \iota^{3/2} + H h_\star^3 S C^{\mathrm{partial}} \iota^2}{n}$$

$$+ \sqrt{\frac{h_\star^4 S C^{\mathrm{partial}} \iota^2}{n}} \cdot H \cdot \varepsilon_1 + \frac{H^2 h_\star^2 S C^{\mathrm{partial}} \iota^{3/2} + H h_\star^2 S C^{\mathrm{partial}} \iota^2}{n}\Bigg) + \varepsilon_0$$

$$\leq C\Bigg(\sqrt{\frac{H^2 h_\star S C^{\mathrm{partial}} \iota^2}{n}} + \frac{H^2 h_\star S C^{\mathrm{partial}} \iota^{3/2} + H h_\star^3 S C^{\mathrm{partial}} \iota^2}{n}$$

$$+ \sqrt{\frac{H^2 h_\star^4 S C^{\mathrm{partial}} \iota^2}{n}} \cdot \Bigg(\sqrt{\frac{H^2 h_\star^3 S C^{\mathrm{partial}} \iota^2}{n}} + \varepsilon_0\Bigg) + \frac{H^2 h_\star^2 S C^{\mathrm{partial}} \iota^{3/2} + H h_\star^2 S C^{\mathrm{partial}} \iota^2}{n}\Bigg) + \varepsilon_0$$

$$=: \varepsilon_2.$$

We first let $\varepsilon_0 \leq \varepsilon/2$ which requires

$$n \geq N_0 := C\big[(H - h_\star)^3 S A \iota^3 (C^{\mathrm{partial}})^2 + (H - h_\star)^3 S^2 A \iota^3/\varepsilon\big].$$

Then to let the rest of the terms above be also bounded by $\varepsilon/2$, a sufficient condition is

$$n \geq N_2 := C\Bigg[\frac{H^2 h_\star S C^{\mathrm{partial}} \iota^2}{\varepsilon^2} + \frac{H^2 h_\star^{3.5} S C^{\mathrm{partial}} \iota^2}{\varepsilon} + H^2 h_\star^4 S C^{\mathrm{partial}} \iota^2\Bigg].$$

Combined, this shows that $\max_{1 \leq h \leq h_\star} \sum_{s \in \mathcal{S}} d^{\pi_\star}_h(s)\Big(V^\star_h(s) - \widehat{V}_h(s)\Big) \leq \varepsilon$ if

$$n \geq C\Bigg(\frac{H^2 h_\star S C^{\mathrm{partial}} \iota^2 + (H - h_\star)^3 S A (C^{\mathrm{partial}})^2 \iota^3}{\varepsilon^2}$$

$$+ \frac{H^2 h_\star^{3.5} S C^{\text{partial}} \iota^2 + (H - h_\star)^3 S^2 A C^{\text{partial}} \iota^3}{\varepsilon} + H^2 h_\star^4 S C^{\text{partial}} \iota^2 \Bigg)$$

$$\geq \max \{N_0, N_2\}.$$

Further, when $\varepsilon \leq \min \{h_\star^{-2.5}, C^{\text{partial}}/S\}$, the $\varepsilon^{-2}$ term above dominates and thus a sufficient condition for the above is

$$n \geq \widetilde{O}\left(\frac{H^2 h_\star S C^{\text{partial}} + (H - h_\star)^3 S A (C^{\text{partial}})^2}{\varepsilon^2}\right). \tag{17}$$

**Guarantees for output policy**   We also show that the final output policy $\widehat{\pi}$ of Algorithm 2 also satisfies $V_1^\star(s_1) - V_1^{\widehat{\pi}}(s_1) \leq \varepsilon$ building on the above guarantee for $\widehat{V}$. First, at step $h_\star + 1$, as $\widehat{\pi}_{(h_\star+1):H} = \widehat{\pi}_{(h_\star+1):H}^{\text{UCB}}$, by Lemma E.1(b) we have for all $s \in \mathcal{S}$ that

$$V_{h_\star+1}^{\widehat{\pi}}(s) \geq \underline{V}_{h_\star+1}(s) = \widehat{V}_{h_\star+1}(s).$$

Second, using this as a base step for the induction argument in C.2, we get that $V_h^{\widehat{\pi}}(s) \geq \widehat{V}_h(s)$ for all $h \in [h_\star]$ and $s$. In particular, at $h = 1$ we have $V_1^{\widehat{\pi}}(s_1) \geq \widehat{V}_1(s_1)$. Therefore

$$V_1^\star(s_1) - V_1^{\widehat{\pi}}(s_1) \leq V_1^\star(s_1) - \widehat{V}_1(s_1) = \sum_{s \in \mathcal{S}} d^{\pi_\star}(s)\left(V_1^\star(s) - \widehat{V}_1(s)\right) \leq \varepsilon.$$

This shows the desired near-optimality guarantee for $\widehat{\pi}$ whenever $\varepsilon \leq \min \{h_\star^{-2.5}, C^{\text{partial}}/S\}$ and the number of episodes $n$ satisfies (17). This proves Theorem 4. $\qquad\square$