# OpenReview forum: "Policy Finetuning: Bridging Sample-Efficient Offline and Online Reinforcement Learning"
_NeurIPS.cc/2021/Conference — NeurIPS 2021 Poster_

### Official Review · Reviewer_rtpe · 2021-07-07

**Rating:** 7
**Confidence:** 3

**Summary:**

The paper proposes a theoretical analysis of off-policy and on-policy RL algorithms, and proposes a new hybrid algorithm that learns a policy in the following way:

- The beginning of episodes, time-steps 1 to h*, is used to learn offline, from samples collected by a behavior policy.
- The end of episodes, time-steps h* to the horizon, is used to learn online by exploring in the environment

The paper proves that there are families of environments and behavior policies for which the proposed algorithm has a better sample-efficiency than either purely online or purely offline approaches.

**Limitations And Societal Impact:**

There was no societal impact to evaluate, as the paper is purely theoretical, and uses equations that introduce no human bias.

**Main Review:**

The paper is very dense, but well written, easy to understand, and does a great job at citing and discussing related work. The proposed algorithm seems quite original, as it is unusual to consider "halves" of episodes, with the early time-steps used for something, and the later time-steps used differently. The theoretical approach and the proofs look sound, and the paper is overall well motivated.

A minor question I have is whether Step 1 of Algorithm 2 (the step that executes the behavior policy for some time-steps, before starting to explore with UCB) could store the transitions acquired by the behavior policy in the experience buffer D, later used for offline learning. The idea would be to be a bit more data efficient, and not to throw away the samples generated by $\mu$ in Step 1.

The paper is overall already very dense, and I don't see how more information could be added to it. I would however suggest (maybe for future work) to also have, linked to the research presented in this paper:

- A numerical evaluation of the proposed algorithm, compared to the baselines, in several environments (that can be simple). The objective is to see how the change in theoretical sample-efficiency influences real-world performance.
- A brief description, maybe a paragraph, of how to implement the equations as part of a practical algorithm. Ideas on how to implement the algorithm with function approximation would have high value, even if the resulting implementation would be incompatible with the proofs given in the paper.

I think that the algorithm proposed in this paper, with its splitting of episodes, is truly novel. I would feel bad if this algorithm were to remain un-implemented for decades, because it never leaves the field of theoretical reinforcement learning.

Author response: The authors say that they will discuss avenues for implementing their contribution. I think that it is absolutely crucial to do so, as the authors are the only people in the world truly expert in their contribution. Finding how to implement it may be a bit more challenging than originally expected, and the authors are usually the only people who may eventually be able to do it. I recommend accepting this paper, as it is novel and well-presented, but I also personally recommend that the authors spend a bit of time producing at least a work-in-progress implementation of the algorithm (a fork of stable-baselines3 on Github for instance). I think that if this paper were to be presented at NeurIPS, having a nice QR-Code in the slides, that points to ready-to-use code, would lead to many citations of this paper.

**Time Spent Reviewing:**

2

---

> ### Author Response · Authors · 2021-08-10
> **Authors' response**
>
> We thank the reviewer for the positive feedback and suggestions for improvement.
>
> *****
>
> > A minor question I have is whether Step 1 of Algorithm 2 (the step that executes the behavior policy for some time-steps, before starting to explore with UCB) could store the transitions acquired by the behavior policy in the experience buffer D, later used for offline learning. The idea would be to be a bit more data efficient, and not to throw away the samples generated by $\mu$ in Step 1.
>
> In Algorithm 2, we required stage 1 and stage 2 to use completely independent data for theoretical purposes, so that concentration analyses in stage 2 can go through; such data splitting (into two halves) will not hurt the $O(\cdot)$-style sample complexity as well. However in practice, we agree that reusing the data from stage 1 could be a good idea indeed to make the algorithm more data efficient.
>
> *****
>
> > I would however suggest (maybe for future work) to also have, linked to the research presented in this paper… numerical evaluation of the proposed algorithm… brief description of how to implement the equations as part of a practical algorithm.
>
> Thank you for this suggestion. Regarding the practical implementation, we could add some discussions along the following lines. First, when the environment is a tabular MDP, both our Algorithm 1 and Algorithm 2 are readily implementable. When there is large state/action space and potentially function approximation, we believe our algorithm can be extended, for example, by replacing all the optimistic/pessimistic value iteration steps by DQN-type algorithms with positive/negative bonus functions. We will add more discussions on such practical implementation issues in our revision, and we think an experimental evaluation of such algorithms would be a good direction for future work.

---

### Official Review · Reviewer_TVkX · 2021-07-14

**Rating:** 6
**Confidence:** 3

**Summary:**

This paper studies policy finetuning, a new reinforcement learning setting that allows us to compare and connect sample-efficient online and offline reinforcement learning. We establish sharp upper and lower bounds for policy finetuning under various assumptions on the reference policy. Our bounds show that the optimal policy finetuning algorithm is either offline reduction or a purely online algorithm in the specific setting where the reference policy satisfies single-policy concentrability, and we also show that a hybrid online/offline algorithm can be advantageous over both in more relaxed settings. Many directions could be of interest for future research, such as alternative assumptions on the reference policy, or policy finetuning with function approximation.

This paper investigates theoretical aspects of policy fine tuning where the goal is to find a near optimal policy when a reference policy is given. They study this problem under two assumptions:

1- Reference policy \mu satisfies single-policy concentrability.
2- Reference policy \mu satisfies concentrability up to a certain time step which is a more relaxed than the first assumption.

Under these assumptions for the first time sharp upper and lower bounds for policy finetuning is stablished.

**Limitations And Societal Impact:**

Authors have not discussed the societal Impact of this work. As with any machine learning algorithm, there are potential societal impacts in how their proposed insights/algorithms is applied.

**Main Review:**

Pros:

1-This paper studies an increasingly important setting for reinforcement learning that can pave the path for solving real world problems via RL.
2-Contributions seems to be original and are the first step towards algorithms for policy finetuning with provably near optimal solutions. Derivations sounds correct as much as i could check.
3-Paper is well written.

Cons:
Paper does not have any experimental results.


**Time Spent Reviewing:**

8

---

> ### Author Response · Authors · 2021-08-10
> **Authors' response**
>
> We thank the reviewer for the thoughtful review.
>
> Regarding the experimental results, our goal in this paper is to focus more on the theoretical side, for example, the design of algorithms and sample-complexity analysis, and we would like to leave the empirical evaluation of our algorithm (e.g. in comparison with other related baselines) as future research.

---

### Official Review · Reviewer_eRNE · 2021-07-19

**Rating:** 7
**Confidence:** 2

**Summary:**

The paper attempts to bridge the gap in theory for online and offline RL. The paper introduces a theoretical study of policy fine-tuning, an online RL setup where the learner has access to policy close to the optimal policy in a certain metric. Then the authors introduce a shart offline reduction algorithm that collects data using this policy and then runs offline RL training on the collected data and prove a converge bound for this algorithm. Then authors establish a lower bound on any policy fine-tuning method. Finally, the authors design a new hybrid offline/online algorithm.

**Limitations And Societal Impact:**

Yes.

**Main Review:**

The paper studies a relevant setup of policy fine-tuning. The fine-tuning setup is really useful for certain RL applications. The proofs seem to be correct but I didn't check the theory thoroughly. The paper is well written and easy to follow.

I have some concerns regarding a lack of any experimental evaluation. But, in general, the paper is well written and studies the problems that are relevant to the RL community. I will be curious to see a follow-up work that addresses similar questions for the function approximation case.

**Time Spent Reviewing:**

8

---

> ### Author Response · Authors · 2021-08-10
> **Authors' response**
>
> Thanks for your thoughtful review and positive feedback.
>
> Regarding the experimental evaluation, our goal in this paper is to focus more on the theoretical side such as the design of algorithms and analyses of sample complexities, and we agree that an experimental evaluation of our algorithm (or other algorithms on policy finetuning) would be an interesting direction for future work.

---

### Official Review · Reviewer_gXvU · 2021-07-19

**Rating:** 6
**Confidence:** 3

**Summary:**

This paper provides a theoretical framework for policy finetuning in episodic finite-horizon MDPs, where the agent can interact with the environment online and also has access to offline data from a reference policy. The authors develop/provides three main results:

(1) A fully offline algorithm based on finite-horizon VI-LCB (Rashidinejad et al.'21) and improving its near-optimal sample complexity by the order of O(H^2) in polylogarithmic time (H being the horizon length). This is done by (a) replacing Hoeffding bonus in VI-LCB with Bernstein bonus and (b) adopting the reference-advantage decomposition technique of (Zhang et al'20)

(2) Under concentrability assumption (boundedness of state-action importance weights w.r.t some optimal policy), the authors argue that there is no finetuning algorithm with better sample complexity than the extreme cases(purely online or the proposed offline algorithm) for some MDP and reference policy. This establishes a lower bound for finetuning.

(3) A partial concentrability assumption (reference policy is only guaranteed to satisfy concentrability up to a certain step) and a two-satge hybrid algorithm based on Nash-VI algorithm(Liu, et.al'20) and truncated version of their proposed offline algorithm.

**Limitations And Societal Impact:**

They have addressed theoretical limitations to some extent

**Main Review:**

The paper is well written. Assumptions and discussions are fairly easy to read.  The authors do a good job motivating the problem and provide a nice formulation of policy finetunning. The hybrid algorithm and partial concentrability assumption are also interesting.

-In the hybrid case(theorem 4), how much the sample complexity depends on the choice of algorithm in the first stage? I was wondering if there is any other choice other than UCBVI-UPLOW that could be used in the algorithm?

line 69: setting *is* equivalent


**Time Spent Reviewing:**

6

---

> ### Author Response · Authors · 2021-08-10
> **Authors' response**
>
> We thank the reviewer for the positive feedback and insightful questions.
>
> *****
>
> > In the hybrid case(theorem 4), how much the sample complexity depends on the choice of algorithm in the first stage?
>
> In Theorem 4, $(H-h_\star)^3SA(C^{\rm partial})^2 / \epsilon^2$ is the part (within the sample complexity) that depends on the first stage (online exploration for step $h_\star+1$ to $H$). Within this term, $(H-h_\star)^3SA / \epsilon^2$ is the part that depends on the quality of the online RL algorithm (our UCBVI-UPLOW achieves this; other suboptimal online RL algorithms may achieve potentially worse rates), and we pay $(C^{\rm partial})^2$ additionally because we use the reference policy $\mu$ to collect data during the first $h_\star$ steps.
>
> *****
>
> > I was wondering if there is any other choice other than UCBVI-UPLOW that could be used in the algorithm?
>
> We could use any online RL algorithm that can efficiently find a near-optimal policy as well as outputting a certified *lower* estimate of the value function of that policy. For example, the Policy Certificates algorithm of Dann et al. 2019 [12] also works here (see Line 324 - 330 for discussions about this). We remark though that vanilla online RL algorithms that only provide *upper* estimates of the value function will not work here without further modifications.

---

### Decision · Program_Chairs · 2021-09-27

**Decision:**

Accept (Poster)

**Comment:**

The reviewers all agreed that this paper provides important theoretical contributions to an important area: sample efficient online learning (fine-tuning), using first offline training. The primary concern is about the practicality of the algorithm and the lack of empirical validation. As a paper focused on theory, however, this is not a concern; such work can naturally be done as a follow-up. As suggested by a reviewer, I highly encourage the authors to use the additional space in the final version to discuss how such an algorithm could be implemented, to make it easier to follow-up on this work.